# Meta-Adaptive Prompt Distillation for Few-Shot Visual Question Answering

**Akash Gupta, Amos Storkey, Mirella Lapata**
School of Informatics, University of Edinburgh
{akash.gupta,a.storkey}@ed.ac.uk, mlap@inf.ed.ac.uk

## Abstract

Large Multimodal Models (LMMs) often rely on in-context learning (ICL) to perform new visual question answering (VQA) tasks with minimal supervision. However, ICL performance, especially in smaller LMMs, does not always improve monotonically when increasing the number of examples. We hypothesize that this happens because the LMM is overwhelmed by extraneous information in the image embeddings that is irrelevant to the downstream task. To address this, we propose a meta-learning approach that induces few-shot capabilities in LMMs through a fixed set of soft prompts distilled from task-relevant visual features, which are adapted at test time using a small number of examples. We facilitate this distillation through an attention-mapper module that can be easily integrated with any LMM architecture and is jointly learned with soft prompts. Evaluation on the VL-ICL Bench shows that our method successfully achieves task adaptation in low-data regimes with just a few gradient steps, outperforming ICL by 21.2%. Comparisons with parameter-efficient finetuning methods demonstrate that meta-learning further enhances this adaptation by 7.7% for various VQA tasks.[1]

## 1 Introduction

Humans have the remarkable ability to quickly learn new tasks in multimodal environments with just a few trial-and-error attempts. Extensive research in cognitive science suggests that this ability arises from learning hierarchical abstractions and maintaining shared structural priors across related tasks based on past experiences (Griffiths et al., 2019; Finn, 2018; Kirsch & Schmidhuber, 2022). Drawing on this prior knowledge enables rapid learning in new situations and reduces the need for large amounts of task-specific demonstrations (Finn et al., 2017).

Large Multimodal Models (LMMs) are able to perform a multitude of tasks ranging from reasoning to fine-grained image understanding and visual question answering (Liu et al., 2024; Li et al., 2023a; Laurençon et al., 2024). They are typically built on top of a base Large Language Model (LLM) by supplementing it with a vision encoder and a connecting module that acts as a bridge for different modalities to interact. When (pre)trained at sufficient scale and finetuned on a wide range of multimodal tasks (with natural language instructions), LMMs can learn *new* tasks by virtue of in-context learning (ICL), i.e., by being prompted with a few input-output examples, without requiring any updates to model parameters (Zhao et al., 2024; Zong et al., 2025; Coda-Forno et al., 2023). Although the training-free nature of ICL has led to its rapid adoption across tasks and domains, its underlying mechanism remains ill-understood (Hendel et al., 2023; Huang et al., 2024) and its empirical behaviour can be inconsistent.

Zong et al. (2025) demonstrate that ICL is most effective for large-scale LMMs (~72B parameters), while smaller models (≤7B parameters) struggle with increasing in-context examples and their performance either plateaus or deteriorates, even when extending the context length or giving detailed instructions. They attribute this limitation to the fact that smaller models struggle with the large number of image tokens in long sequences. They become confused and perform the task haphazardly or default to their parametric knowledge, effectively ignoring the in-context examples. Figure 1 shows a failure case from the Fast Open-Ended MiniImageNet dataset (Tsimpoukelli et al., 2021), using

---

[1] We release our code here - https://github.com/akashgupta97/MAPD

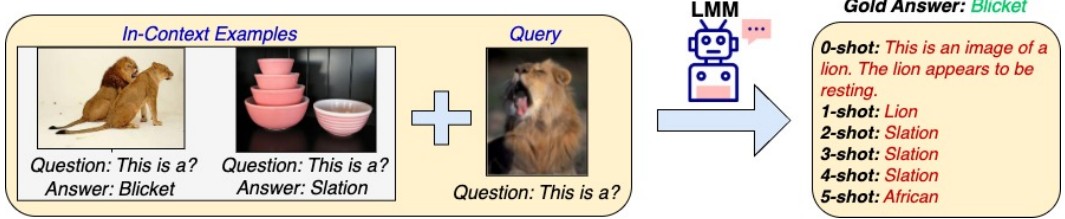

Figure 1: Failure case of LLaVA-OneVision-7B (Li et al., 2025) on an example from the Fast Open-Ended MiniImageNet classification task (Tsimpoukelli et al., 2021). When no in-context examples are provided (0-shot), the model generates a generic description of the image. As more examples (shots) are added, it begins to learn the answer format (single word), but still fails to grasp the task, producing incorrect or irrelevant predictions. We only show the in-context examples (left) for 2-way 1-shot setting for the sake of brevity but provide model predictions (in red) for up to 5 shots.

LLaVA-OneVision-7B (Li et al., 2025). The task is framed in a 2-way N-shot format where a *support* set with N labeled examples of two classes is provided. The model uses ICL with the support set to classify new *query* examples from the two classes. Without any support set or in-context examples (0-shot), the model outputs a generic description about the image based on parametric knowledge and ultimately fails to answer correctly, despite being prompted with a few examples.

Building on this observation, we hypothesize that effective few-shot adaptation at test time may be compromised by the information added by the image embeddings. Figure 2 compares Image-to-Text (I2T, red) and Text-to-Text (T2T, blue) performance of LLaVA-OneVision-7B on the Operator Induction and CLEVR Count Induction tasks (see Appendix A.2.2). The results reveal a significant performance gap: T2T ICL consistently outperforms I2T and improves monotonically with additional shots. Moreover, adding detailed task instructions to I2T (green; see Appendix A.2.1) actually degrades performance, suggesting that naively increasing the number of image embeddings in context impairs the model's inherent ICL ability. While a set of more precise image embeddings would be preferable, their continuous nature makes it challenging to distill task-specific information from them. As an alternative, we propose to *learn* a fixed set of *new* embeddings that can be easily finetuned at test time.

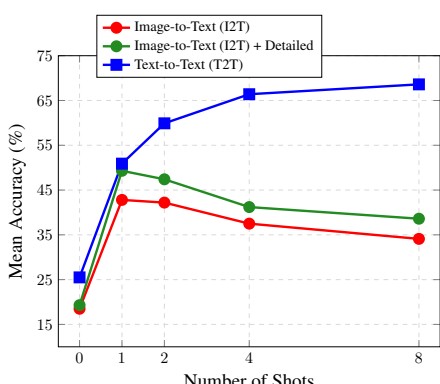

Figure 2: I2T and T2T performance with LLaVA-OneVision-7B on Operator Induction and CLEVR Count Induction tasks.

This idea of task adaptation has gained significant traction in the literature through *prompt tuning* (Lester et al., 2021) which finetunes a set of continuous *soft* prompts while keeping the underlying language model frozen; the prompts are prepended in the context at test time, effectively steering the model toward the desired task. Our approach learns new tasks using soft prompts that receive task information from the LLM in the form of loss gradients during finetuning. These gradients update the soft prompts which when fused with the image embeddings are able to distill relevant features from them. To facilitate this fusion, we propose an attention-mapper that uses a multi-head attention (Vaswani et al., 2017) architecture for extracting relevant task-specific image information and can be substituted in the projection layer of any LMM architecture.

Our approach relies on rapidly adapting to new tasks at test time using only a few examples, which is not addressed by traditional finetuning methods. Prior work (Finn et al., 2017; Ravi & Larochelle, 2017) addressed this challenge by training a meta-learner that can infer an optimal learning strategy for a new task after being exposed to a distribution of tasks. We apply this procedure to our multimodal prompt distillation setting by employing the widely known MAML algorithm (Finn et al., 2017) and use its lightweight first-order approximation to train the attention-mapper and soft prompts. We focus on visual question answering (VQA; Antol et al. 2015; see example in Figure 1), a general-purpose

task often used to evaluate the image understanding capabilities of LMMs, and demonstrate the benefits of MAML training applied to LMM architectures. Our contributions are as follows:

- We introduce MAPD (**M**eta-**A**daptive **P**rompt **D**istillation), an alternative to in-context learning that meta-learns a fixed set of soft prompts within large multimodal models (LMMs) via distillation. MAPD enables adaptation to new tasks with a few examples using a few gradient updates at test time, and consistently improves performance as the number of shots increases. To our knowledge, this is the first exploration of meta-learned prompt distillation for cross-task generalization in LMMs under low-data settings.
- We propose a flexible attention-mapper module, inspired by Najdenkoska et al. (2023), that exploits all patch features from the vision encoder and can be readily incorporated into the projection layer of any LMM architecture. It is trained jointly with *soft* prompts and can be efficiently adapted at test-time to distill task-specific visual information.
- Extensive evaluation on VL-ICL Bench.[2] (Zong et al., 2025), a diverse benchmark for image perception and mathematical reasoning, demonstrates that our approach outperforms ICL and several other prompt distillation and parameter-efficient finetuning methods.

## 2 RELATED WORK

Our approach, Meta-Adaptive Prompt Distillation (MAPD), builds upon several existing research areas including few-shot learning, prompt tuning and test-time adaptation.

**Multimodal Few-shot Learning** Learning from a few examples has been a long-standing goal in machine learning. Early work by Vinyals et al. (2016) introduced Matching Networks for one-shot image-to-text classification. This approach leverages a support set of labeled images to classify an unlabeled query image, laying the foundation for few-shot learning in vision tasks. With the advent of large language models (LLMs) and large multimodal models (LMMs; Alayrac et al. 2022; Zhao et al. 2024), in-context learning (ICL; Zhao et al. 2024; Lester et al. 2021) has emerged as a popular method for few-shot adaptation. ICL involves providing a few input-output examples directly in the model's prompt without updating its parameters. While this is a computationally inexpensive method, its performance for LMMs can be inconsistent (Zong et al., 2025) and may even degrade as more examples are added, particularly in smaller models.

**Learning with Prompts** Another widely adopted approach for adapting models to task-specific data is prompt optmization. Wang et al. (2022) explored this idea with small language models ($\sim 0.1$M) such as BERT for text classification. Subsequent work (Hou et al., 2022) introduced soft prompts to overcome the limitations of optimizing over discrete vocabulary tokens. Khattak et al. (2023) further proposed PromptSRC for CLIP-based vision-language encoders, mitigating overfitting of soft prompts. While these methods perform well on classification tasks, their extension to LLM-based architectures and problems such as question-answering and mathematical reasoning remains limited.

**Test-Time Adaptation** These methods aim to dynamically adapt models during inference on test examples, that may have distributional differences from the training data. This adaptation can either involve training of model parameters (Hardt & Sun, 2024) or can be entirely training free (Karmanov et al., 2024). Additionally, previous work (Hu et al., 2025) has taken advantage of prompt tuning and other PEFT methods such as LoRA (Hu et al., 2022) to resolve catastrophic forgetting issues during test time training and achieve state-of-the-art performance. Shu et al. (2022) propose Test-time Prompt Tuning (TPT), a method that adapts vision-language models for zero-shot classification by tuning soft prompts on image augmentations. Previous work (Najdenkoska et al., 2023; Li et al., 2023b) has also explored meta-learning of soft prompts for small models and a limited range of vision-language tasks such as fast-concept binding.

We extend upon this idea to provide an alternative for few-shot adaptation in LMMs. Specifically, we design a meta-learning procedure, namely MAPD, to learn soft prompts that can distill task-relevant visual features from image embeddings and can be rapidly adapted at test time for a variety of new tasks using a few examples. Najdenkoska et al. (2023) rely on a single [CLS] token from CLIP's vision encoder, which limits the attention-mapper's capacity. We instead use the complete set of hidden patch features, enabling the attention-mapper to encode fine-grained visual information for

---

[2]We only focus on single-image few-shot VQA tasks and leave the multi-image scenario for future work.

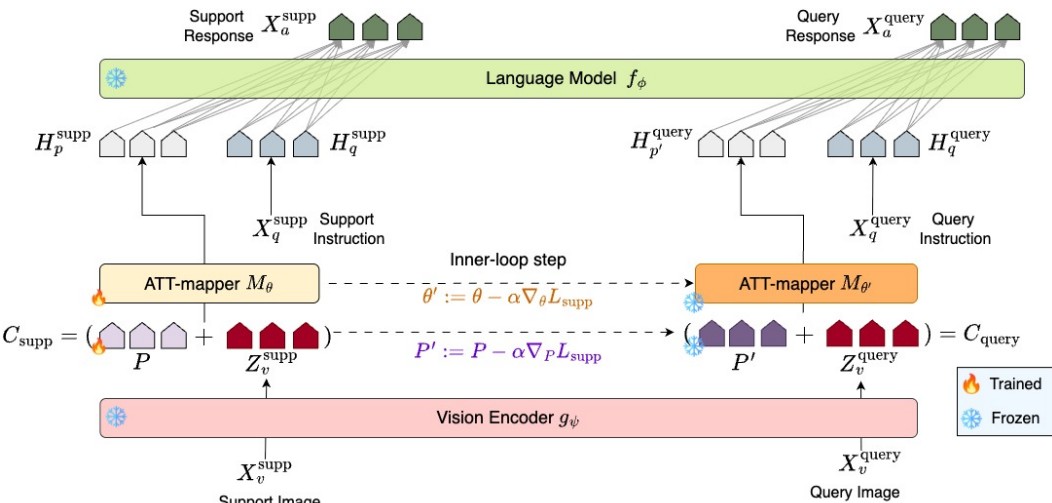

Figure 3: Our proposed MAPD framework based on LLaVA v1.5-7B (Liu et al., 2024): image embeddings are distilled into soft prompts $P$ during instruction finetuning. The support set $(X_v^{\text{supp}}, X_q^{\text{supp}}, X_a^{\text{supp}})$ is processed initially to the obtain loss value $L_{\text{supp}}$ which is used in the inner-loop to obtain task-specific parameters $\{\theta', P'\}$. Next, the query set $(X_v^{\text{query}}, X_q^{\text{query}}, X_a^{\text{query}})$ is used to calculate the query loss for the outer-loop meta-parameter optimization $\{\theta, P\}$.

distillation into soft prompts. We show that MAPD can be applied to any LMM architecture and achieves state-of-the-art performance on visual question answering tasks (Antol et al., 2015).

## 3 PROBLEM FORMULATION

### 3.1 FEW-SHOT VISUAL QUESTION ANSWERING

Visual Question Answering (VQA; Antol et al. 2015) is a key task for evaluating the ability of vision-language models to understand images by accurately responding to questions about various aspects of visual content. These questions can vary widely, ranging from descriptions of objects inside bounding boxes (Krishna et al., 2017) to solving high-school geometry problems (Gao et al., 2025), but are mostly grounded in the visual information present in the image.

In VQA, we typically have a dataset $\mathcal{D} = \{(X_v^i, X_q^i, X_a^i)\}_{i=1}^{|\mathcal{D}|}$ where $X_v \in \mathcal{I}$, $X_q \in \mathcal{Q}$ and $X_a \in \mathcal{A}$, and $\mathcal{I}$ is the set of all images, $\mathcal{Q}$ the set of all questions, and $\mathcal{A}$ the set of all answers. Our goal is to learn a function $f_\theta$ parametrized by $\theta$, that maximizes the likelihood of the answer given the image and the question, $\prod_{i=1}^{|\mathcal{D}|} p_\theta(X_a^i|X_v^i, X_q^i)$. Following the standard train-test paradigm in deep learning, we evaluate whether $f_\theta$ generalizes well by dividing dataset $D$ into $(D^{\text{train}}, D^{\text{test}})$ such that maximizing the above likelihood on $D^{\text{train}}$ also maximizes the likelihood of answer on $D^{\text{test}}$. A common assumption is that the size of $D^{\text{train}}$ is large enough so that function $f_\theta$ does not overfit on $D^{\text{train}}$. In the context of *few-shot* VQA, we treat the in-context examples (or shots) given to an LMM during ICL as $D^{\text{train}}$. Since the examples in $D^{\text{train}}$ are limited (as few as 1-shot), it becomes harder to avoid overfitting while training and still perform well on $D^{\text{test}}$. We conceptualize this problem as one of learning about an underlying task represented by $D^{\text{train}}$ and adopt meta-learning (Finn et al., 2017) which exploits the shared structure across a distribution of tasks to learn a prior over model parameters, thereby enabling stable transfer to new tasks with limited data. In the following, we describe how we enforce this prior over parameters through the curation of *meta-tasks* containing few-shots. A sketch of our model architecture and training procedure is shown in Figure 3.

### 3.2 IMPROVING TASK UNDERSTANDING WITH META-TASKS

The core idea of optimization-based meta-learning is to learn a good initialization of parameters, which when finetuned on a specific task, enables stable transfer for that task with a few gradient

steps (Finn et al., 2017). To promote this capability, training involves processing batches of few-shot datasets that represent an underlying task. We refer to these few-shot datasets as *meta-tasks* and propose to create them from our finetuning data mixture based on the original LLaVA datasets. We provide details of our specific data mixture in Appendix A.1.1.

More formally, let $p(\mathcal{D})$ denote our data mixture. We create meta-task $T^j$ by randomly sampling a fixed subset of VQA examples (image, question, answer triplets) from dataset $D^i \sim p(\mathcal{D})$ and partitioning the examples further into support and query sets $T^j = \{D^{\text{supp}}, D^{\text{query}}\}$. To be consistent with the notation introduced in Section 3.1, we treat the support set as $D^{\text{supp}} \equiv D^{\text{train}}$ and the query set as $D^{\text{query}} \equiv D^{\text{test}}$. We continue this process until all samples from $D^i$ have been assigned to at least one meta-task. This meta-task construction is performed for *each dataset* in $p(\mathcal{D})$, resulting in meta-task distribution $p(\mathcal{T}^{\text{meta}})$. We now describe our model architecture designed to process these meta-tasks. Further details on the number and composition of meta-tasks for training and evaluation are provided in Appendix A.1.2.

### 3.3 Model Architecture

We design our LMM architecture (Figure 3) based on the visual instruction tuning framework of LLaVA v1.5[3] (Liu et al., 2024) and further describe our modifications for incorporating the attention-mapper. For clarity, we omit the distinction between support and query sets in this section as both are processed in the same manner. As shown in Figure 3, the model consists of a pretrained CLIP ViT-L/14 visual encoder ($g_\psi$) with an aspect ratio of 336px; for an input image $X_v$, the encoder gives us hidden visual features $Z_v$ which are then passed to the projection layer that consists of an attention-mapper $M_\theta$ responsible for extracting useful features from $Z_v$.

**Attention Mapper**   We re-design the projection layer of LLaVA v1.5 to include soft prompts $P$ by introducing an attention-mapper $M_\theta$ for improved *task-specific* feature extraction. Specifically, we prepend $Z_v$ with a set of $m$ learnable prompt tokens $P$ to obtain a sequence $C = (P, Z_v)$ which is then passed to the attention-mapper (see Figure 3). Both prompt tokens $P$ and weights $\theta$ are initialized with Xavier Uniform initialization (Glorot & Bengio, 2010). We define the mapper as:

$$H_{p+v} = M_\theta(Q, K, V) = \sigma(QK^T) * V \tag{1}$$

where the query is $Q = M_\theta^q \cdot C$ ; , the key is $K = M_\theta^k \cdot C$ ; , the value is $V = M_\theta^v \cdot C$, and their corresponding matrices are $\{M_\theta^q, M_\theta^k, M_\theta^v\}$. The mapper computes the dot product of the query and key vectors which are then passed to a softmax function to compute activation scores for every feature in vector $V$. Finally, we extract the first $m$ embeddings corresponding to the learnable prompt tokens $V$ from the set $H_{p+v}$ that correspond to the task-specific image embeddings $H_p$. These are now passed to the LLM ($f_\phi$) as prompts for further processing. We denote the trainable parameters for the attention-mapper with $\theta_p = \{\theta, P\}$.

**Language Model**   The quality of the learned prompts highly depends on the underlying language model. We update the LLM of LLaVA v1.5 with the state-of-the-art Qwen2.5-7B-Instruct LLM, which has demonstrated strong performance on complex tasks such as mathematical reasoning and coding and supports the generation of up to 8K tokens. The LLM ($f_\phi$) receives the concatenated sequence of image and text tokens to generate the answer $X_a = f_\phi([H_p, H_q])$. In this pipeline, only the attention mapper parameters $\theta_p$ are trained, making our approach parameter-efficient for cross-task generalization. The number of trainable parameters is approximately 24M (see Appendix A.1.3 for hyperparameters). The training objective maximizes the likelihood function, $p_{\theta_p}(X_a | X_v, X_q)$, parametrized by $\theta_p$, where $X_a$ is the answer, $X_v$ is the image, and $X_q$ is the question. For clarity, we refer to this model, namely LLaVA-ATT-Qwen2.5 7B, as our base LMM in the following sections.

### 3.4 Model Training

We train the attention mapper parameters to learn image-conditioned soft prompts in two stages following a curriculum learning procedure similar to LLaVA v1.5 (Liu et al., 2023). In the first-stage, which is aimed at feature alignment, the attention-mapper is pretrained on the LCS-558K subset

---

[3]We adopt LLaVA v1.5 due to its simplicity and publicly available training code and datasets (Section A.1.1). This prevents mixing between training and test datasets and enables evaluation over unseen tasks. We demonstrate in Section 4.3 that our method can be easily applied to other LMM architectures.

of the LAION/CC/SBU dataset filtered with a more balanced concept coverage (Liu et al., 2023). Further details on pretraining are mentioned in Appendix A.1.4. In the second stage, which aims to distill task-specific image features into prompts $H_p$, the attention-mapper parameters $\theta_p$ are finetuned on diverse task-specific instructions. We describe our MAML-based finetuning procedure below and also introduce alternative methods which we compare against in our experiments.

### 3.4.1 LEARNING TO DISTILL PROMPTS WITH FIRST-ORDER META LEARNING

Our prompt distillation procedure, MAPD, uses the model-agnostic first-order approximation of MAML (Finn et al., 2017) which aims to learn a robust initialization of meta-parameters that enable efficient adaptation to new tasks with just a few gradient updates. We borrow the implementation of Antoniou et al. (2019) and use their first-order version and (learnable) per-step learning rates ($\alpha$) to further optimize the training process. We sample a batch $B$ of meta-tasks from $p(\mathcal{T}^{\text{meta}})$ and use the support set of each task to convert $\theta_p$ into task specific parameters $\theta'_p$ with a few gradient steps. Equations (2) and (3) show a *single* step of this inner loop:

$$L_{\theta_p}^{\text{supp}} = \frac{-1}{|D^{\text{supp}}|} \sum_{i=1}^{|D^{\text{supp}}|} \log(p_{\theta_p}(X_a^i | X_v^i, X_q^i)) \quad (2) \qquad\qquad \theta'_p = \theta_p - \alpha \nabla_{\theta_p} L_{\theta_p}^{\text{supp}} \qquad (3)$$

The *outer* loop involves optimizing the meta-parameters which in our case are the original attention-mapper parameters $\theta_p$ on the query set using the task-specific parameters $\theta'_p$:

$$L_{\theta'_p}^{\text{query}} = \frac{-1}{|D^{\text{query}}|} \sum_{i=1}^{|D^{\text{query}}|} \log(p_{\theta'_p}(X_a^i | X_v^i, X_q^i)) \quad (4) \qquad \theta_p := \theta_p - \beta \sum_{j=1}^{|B|} \nabla_{\theta'_{p,j}} L_{\theta'_{p,j}}^{\text{query}} \qquad (5)$$

Equation (5) is the first-order approximation of the meta-update in MAML (Finn et al., 2017) that treats the gradient of $\theta'_{p,j}$ w.r.t. $\theta_p$ for a meta task as a constant. This approximation avoids backpropagating through the entire computation graph of the inner loop and avoids the Hessian-vector product estimation of the query loss. This saves huge GPU memory while still approximating a gradient in the same direction as the true MAML gradient (Weng, 2018). We provide a sketch of MAPD training in Figure 3 and a more detailed algorithm in Appendix A.1.5 as Algorithm 1

### 3.4.2 ALTERNATIVE METHODS FOR PROMPT DISTILLATION

We also implement other prompt distillation methods based on our model architecture to compare their performance with MAPD on few-shot VQA tasks. We provide a more formal description of these methods below, highlighting important differences from our framework.

**Multi-Task Prompt Distillation**  We define a multi-task baseline where we eliminate the bi-level optimization of MAPD. Specifically, at each iteration, we sample a batch of meta-tasks from $p(\mathcal{T}^{\text{meta}})$ and optimize the following loss per task:

$$L_{\theta_p} = \frac{-1}{N} \sum_{i=1}^{N} \log(p_{\theta_p}(X_a^i | X_v^i, X_q^i)) \qquad (6)$$

such that $N = |D^{\text{supp}}| + |D^{\text{query}}|$. This loss is accumulated across the entire batch of meta-tasks used to update $\theta_p$. We refer to this baseline as Multi-Task$^{\text{PD}}$.

**In-Context Prompt Distillation**  Previous work (Chen et al., 2022; Min et al., 2022) suggests it is possible to meta-learn task information by reducing the bi-level optimization of MAML to a sequence prediction problem over in-context examples with the help of pretrained LLMs. We develop a method called In-Context$^{\text{PD}}$, where we concatenate the support set with each query example in a meta-task, and optimize the following loss function to distill this task information from LLMs into soft prompts:

$$L_{\theta_p} = \frac{-1}{|D^{\text{query}}|} \sum_{i=1}^{|D^{\text{query}}|} \log(p_{\theta_p}(X_a^i | X_v^i, X_q^i, D^{\text{supp}})) \qquad (7)$$

**Methods without Meta-tasks**  To further understand the benefit of curating meta-tasks (see Section 3.2), we compare with the original finetuning procedure of LLaVA-v1.5 7B but only train $\theta_p$

without any meta-tasks for fair comparison. We refer to this method as NoMeta-task[PD] in subsequent sections. We also compare with model averaging, which is computationally efficient and has been shown to increase performance on out-of-distribution datasets (Choshen et al., 2022; Wortsman et al., 2022).We separately finetune the attention-mapper on each dataset $D^i \sim p(\mathcal{D})$, and take an average of all dataset-specific parameters $\theta_p^i$ weighted by their corresponding dataset size ratios:

$$\theta_p^{\mathrm{avg}} = \sum_{i=1}^{|\mathcal{D}|} \theta_p^i \cdot w^i \tag{8}$$

where $w^i = |D^i| \, / \, |\mathcal{D}|$. We refer to this baseline as Model-Avg[PD] in subsequent sections.

## 3.5 Test-Time Adaptation

After learning optimal parameters with MAPD (and alternative distillation strategies), we adapt the attention-mapper to a new test task by fine-tuning for $K$ gradient steps. We empirically find that $K \leq 30$ is sufficient for all prompt distillation methods to converge, which we attribute to our adaptation procedure training only 24M parameters over a few examples. We further explain how this value is selected in Appendix A.2.3. Concretely, given K $\leq$ 30 steps, we perform task-specific finetuning of the parameters $\theta_p$ on the support set $D_{\mathrm{test}}^{\mathrm{supp}}$ of test task $T_{\mathrm{test}}^j$, using the inner-loop optimizer mentioned in equation 3. We then evaluate model performance on the query set $D_{\mathrm{test}}^{\mathrm{query}}$ for that task.

## 4 Experimental Results

### 4.1 Evaluation Datasets

For evaluation purposes, our test datasets follow the same structure as the meta-tasks introduced in Section 3.2, with support and query examples. We use the recently introduced VL-ICL benchmark (Zong et al., 2025), designed to test the ICL capabilities of LMMs on various tasks like fast concept binding, multimodal reasoning, and fine-grained perception. Meta-tasks for testing are created by randomly sampling a support set from the training split of the VL-ICL datasets and a test/query set from their respective testing splits.[4] In line with our training pipeline, which exclusively utilizes datasets containing a single image per example (see Section A.1.1), we focus solely on single image-to-text scenarios, leaving multi-image cases for future work. We report results on four tasks from VL-ICL: a) *Fast Open MiniImageNet (Open-MI)*, where the model must name new objects based on a few examples; b) *Operator Induction*, where the model must solve image tasks of the type $2 \, ? \, 7 =?$ given training examples like $1 \, ? \, 3 = 4$; c) *CLEVR Count Induction*, where the model must count objects that satisfy given attributes like "*shape: sphere*"; and d) *TextOCR*, where the model must transcribe highlighted text contained in an image. We provide more details on these tasks in Appendix A.2.2. The final model performance is calculated as the average across all meta-tasks.

### 4.2 Model Comparisons

Our results are summarized in Table 1, which compares MAPD against alternative prompt distillation methods (see Section 3.4.2) and reports the mean accuracy of up to eight shots. We compare two types of test-time adaptation methods, namely in-context learning (ICL) which prompts the underlying LLM with no distillation of image embeddings and finetuning (FT) with K $\leq$ 30 gradient steps, which are further distinguished based on whether they use meta-tasks during training. Results for individual shots are in Appendix A.2.4; additional results for more shots are in Appendix A.2.11[5].

**Prompt distillation improves task induction in LMMs at test-time.** Our results in Table 1 show that FT adaptation with few-shots (support examples) largely outperforms ICL at test time evaluated over query examples, with an average increase of 21.2% over all datasets. These results highly support our hypothesis that distilling task-specific information from image embeddings to create targeted prompts improves the few-shot capabilities of the underlying LLM (in our case Qwen-2.5-7B-Instruct).

---

[4]We also keep a separate validation set for each VL-ICL dataset (sampled from the training split) to select the best model which we then evaluate on the test (query) set. More details can be found in Section A.2.3

[5]We also provide ICL performance of publicly available models in Appendix A.2.5 for reference.

| Methods | MT | Open-MI | OP_IND | CLEVR | TextOCR |
|---|---|---|---|---|---|
| **TTA with ICL** | | | | | |
| NoMeta-task[PD] | ✗ | $43.8 \pm 0.9$ | $12.1 \pm 0.6$ | $18.0 \pm 0.2$ | $6.8 \pm 0.4$ |
| Model-Avg[PD] | ✗ | $26.6 \pm 0.7$ | $9.2 \pm 0.5$ | $7.6 \pm 0.1$ | $2.8 \pm 0.3$ |
| In-Context[PD] | ✓ | $51.1 \pm 0.9$ | $20.6 \pm 0.8$ | $24.1 \pm 0.2$ | $23.8 \pm 0.3$ |
| Multi-Task[PD] | ✓ | $48.6 \pm 0.9$ | $10.0 \pm 0.6$ | $12.5 \pm 0.1$ | $6.9 \pm 0.4$ |
| MAPD | ✓ | $53.3 \pm 0.9$ | $9.60 \pm 0.5$ | $12.3 \pm 0.1$ | $7.30 \pm 0.4$ |
| **TTA with FT $\leq 30$** | | | | | |
| NoMeta-task[PD] | ✗ | $68.0 \pm 0.8$ | $38.8 \pm 0.6$ | $25.8 \pm 0.2$ | $22.5 \pm 0.3$ |
| Model-Avg[PD] | ✗ | $63.1 \pm 0.8$ | $40.0 \pm 0.6$ | $29.1 \pm 0.2$ | $21.5 \pm 0.3$ |
| In-Context[PD] | ✓ | $64.5 \pm 0.8$ | $30.9 \pm 0.5$ | $30.9 \pm 0.2$ | $18.9 \pm 0.3$ |
| Multi-Task[PD] | ✓ | $74.6 \pm 0.7$ | $45.1 \pm 0.5$ | $29.9 \pm 0.2$ | $22.9 \pm 0.4$ |
| **MAPD** | ✓ | $\mathbf{77.9 \pm 0.7}$ | $\mathbf{47.7 \pm 0.5}$ | $\mathbf{31.4 \pm 0.2}$ | $\mathbf{26.4 \pm 0.5}$ |

Table 1: Evaluation on tasks from the VL-ICL Bench (Zong et al., 2025) with LLaVA-ATT-Qwen2.5 7B as the base LMM. Each method trains 24M attention-mapper parameters. We report the mean accuracy across shots $\{1, 2, 4, 5, 8\}$ with 95% binomial confidence intervals and compare different prompt distillation approaches. TTA:Test-Time Adaptation, FT: Finetuning with $K \leq 30$ gradient steps, ICL: In-Context Learning, MT: Meta-Tasks used (✓) or not (✗) during training. Qualitative results are in Appendix A.2.6 and A.2.9.

Additionally, our results show that finetuning just the attention-mapper parameters only requires a few gradient steps ($K \leq 30$) at test-time to generalize to unseen tasks and does not lead to overfitting over the support examples (Appendix A.2.3). For a one-to-one comparison, we look

| LoRA | Open-MI | OP_IND | CLEVR | TextOCR |
|---|---|---|---|---|
| **TTA with FT $\leq 30$** | | | | |
| All    LLM layers | $55.1 \pm 0.7$ | $13.3 \pm 0.6$ | $15.1 \pm 0.2$ | $10.4 \pm 0.4$ |
| [0-15] LLM layers | $67.3 \pm 0.8$ | $25.5 \pm 0.6$ | $\mathbf{30.0 \pm 0.2}$ | $23.8 \pm 0.3$ |
| [0-15] LLM layers + ATT | $\mathbf{69.1 \pm 0.8}$ | $\mathbf{30.5 \pm 0.5}$ | $28.7 \pm 0.2$ | $\mathbf{24.5 \pm 0.3}$ |

Table 2: LoRA configurations for base LMM and evaluated on the VL-ICL bench. We report the mean accuracy across shots $\{1, 2, 4, 5, 8\}$ with 95% binomial confidence intervals. ATT: Attention-Mapper; TTA: Test-Time Adaptation; FT: Finetuning with $K \leq 30$ gradient steps.

into In-context[PD], which performs better with FT on 3 out of 4 tasks compared to its ICL adaptation and enables prompting the underlying LLM with a fixed set of learned task-specific embeddings.

**Meta-learning and meta-tasks improve few-shot learning.** Table 1 shows that methods using meta-tasks are indeed superior. For ICL-based adaptation, In-Context[PD] performs best, while for FT-based adaptation, our proposed approach, MAPD, achieves the best overall performance across all four datasets at test time. This further suggests that first-order MAML learns the best initialization of attention-mapper parameters $\theta_p$. These parameters are subsequently adapted for a test task with a few gradient steps and few-shot examples to produce a precise set of soft prompts that improves LMM predictions on that task. Our detailed results in Table 11 in Appendix A.2.4 further show that for FT-based adaptation, MAPD is most effective in the 2-shot case for Operator Induction, surpassing Multi-Task[PD] by 10%. Finally, MAPD with FT is the only approach that exhibits strictly monotonic improvements as the number of shots increases, showing better scaling behavior.

**MAPD surpasses other efficient finetuning approaches for few-shot adaptation.** We compare MAPD with LoRA (Hu et al., 2022), a state-of-the-art parameter-efficient finetuning (PEFT) approach. In Table 2, we integrate LoRA in the base LMM in three configurations and evaluate on VL-ICL: (1) naively applying LoRA to all underlying LLM layers (as done in LLaVA v1.5; Liu et al. 2023) increases the number of trainable parameters ($\sim 300M$) and the model finds it difficult to converge within 30 gradient steps at test-time; (2) restricting LoRA to the first 16 LLM layers ($\sim 24M$ parameters) offers better test-time performance; and (3) adding LoRA to the attention-mapper layers further boosts performance as it provides some distillation over the image embeddings before prompting the underlying LLM. Ultimately, MAPD still outperforms the best LoRA configuration

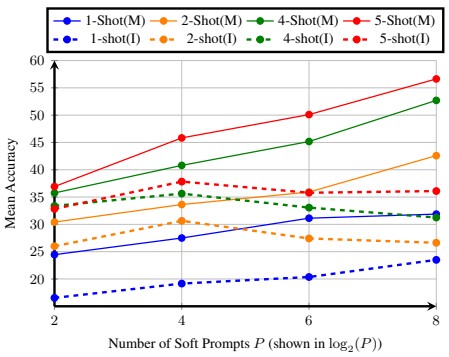 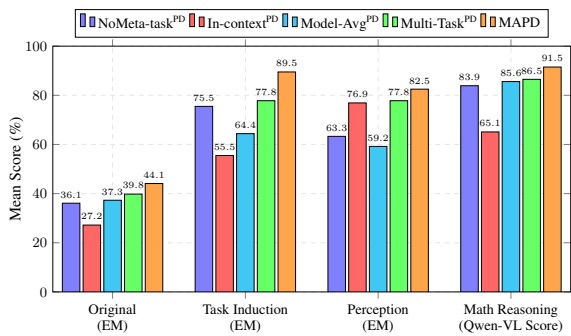

Figure 4: (a) **Left**: Performance comparison between MAPD+FT (M) and In-Context[PD]+ICL (I). Mean Accuracy is computed across all VL-ICL datasets. We consider different prompt token lengths $P = \{4, 16, 64, 256\}$ which are shown in $\log_2(P)$ scale for different shots. (b) **Right**: Performance of different prompt distillation methods on three Operator Induction subtasks: Task Induction, Perception, and Math Reasoning. We report mean exact-match (EM; %) for 1,2 and 8-shots as defined in the VL-ICL Bench (Zong et al., 2025) except for Mathematical Reasoning, which uses mean ratings generated by Qwen-2.5-VL-32B-Instruct. More details can be found in Appendix A.2.9

by an average of 7.7% across all VL-ICL datasets. This demonstrates that MAPD is the best choice for achieving fast test-time adaptation in low-data scenarios. We provide additional LoRA training details in Appendix A.1.4 and further detailed results can be found in Appendix A.2.4.

## 4.3 ABLATION STUDIES AND ANALYSIS

In this section, we present ablation studies across various model architectures and sizes, along with a more in-depth analysis of the benefits of test-time fine-tuning using MAPD. Please refer to appendix for further ablations on testing 1) robustness to image perturbations (Appendix A.2.7) and 2) different few-shot selection strategies (Appendix A.2.8).

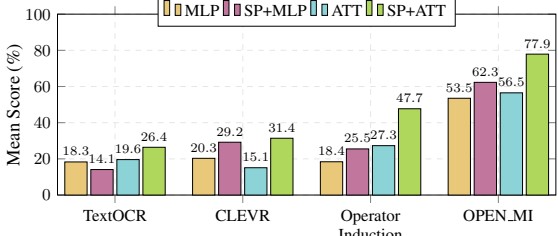

Figure 5: Projection layer architectures in the base LMM. SP: Soft Prompts, ATT: Attention-Mapper, MLP: 2-layer MLP (originally used in LLaVA v1.5).

**What are the benefits of the attention mapper and soft prompts?** In Figure 5, we compare different designs for the projection layer in the base LMM for rapid few-shot learning. We clearly see that MAPD benefits most by incorporating the attention-mapper and soft prompts (SP+ATT). We draw two key conclusions from this experiment: (1) distilling task-relevant information from CLIP embeddings with soft prompts yields substantial improvements, with an average gain of 16.3% across architectures. (2) replacing the 2-layer MLP used in LLaVA v1.5 with an attention mapper leads to an additional average gain of 13.1%, thanks to its inherent weighting mechanism of pairwise similarities over CLIP embeddings.

**How does the number of soft prompts affect performance?** We examine how MAPD's performance changes with the number of soft prompts across varying shot settings for VL-ICL datasets in Figure 4(a). Additionally, we show results of our best ICL approach, In-Context[PD], as a baseline for this comparison. We see that MAPD scales favorably and learns more consistent task information from gradient updates at test time as the number of soft prompts is increased. Furthermore, its marginal improvement per added prompt token is substantially greater when more shots are provided. In contrast, the performance of In-Context[PD] generally deteriorates with more prompts and struggles to jointly attend to more examples and longer prompts. We also investigate this further by analysing the softmax attention values of the underlying LLM for soft prompt embeddings containing image information in Appendix A.2.10. We note that the attention entropy decreases as the context length grows for In-Context[PD]. This highlights its inherent limitation in being unable to attend to all the soft prompts when answering a query, while MAPD consistently attends to all the soft prompts.

| Vision Encoder | LLM | TTA | NoMeta-task$^{PD}$ | Model-Avg$^{PD}$ | In-Context$^{PD}$ | Multi-Task$^{PD}$ | MAPD |
|---|---|---|---|---|---|---|---|
| CLIP ViT-L/14 | Qwen2.5-7B | ICL | $43.8 \pm 0.9$ | $26.6 \pm 0.7$ | $51.1 \pm 0.9$ | $48.6 \pm 0.9$ | $\mathbf{53.3 \pm 0.9}$ |
|  | Instruct | FT$\leq$30 | $68.0 \pm 0.8$ | $63.1 \pm 0.8$ | $64.5 \pm 0.8$ | $74.6 \pm 0.7$ | $\mathbf{77.9 \pm 0.7}$ |
| CLIP ViT-L/14 | Qwen2.5-3B | ICL | $24.3 \pm 0.7$ | $30.5 \pm 0.7$ | $\mathbf{48.3 \pm 0.9}$ | $39.1 \pm 0.7$ | $32.9 \pm 0.7$ |
|  | Instruct | FT$\leq$30 | $56.5 \pm 0.9$ | $66.0 \pm 0.5$ | $47.5 \pm 0.9$ | $61.1 \pm 0.8$ | $\mathbf{67.3 \pm 0.6}$ |
| CLIP ViT-L/14 | Vicuna v1.5-7B | ICL | $20.0 \pm 0.7$ | $26.2 \pm 0.7$ | $46.3 \pm 0.9$ | $29.1 \pm 0.7$ | $\mathbf{49.9 \pm 0.9}$ |
|  |  | FT$\leq$30 | $69.1 \pm 0.8$ | $74.9 \pm 0.4$ | $66.7 \pm 0.8$ | $70.3 \pm 0.7$ | $\mathbf{75.8 \pm 0.4}$ |
| SigLIP-SO400M | Qwen2.5-7B | ICL | $42.6 \pm 0.9$ | $40.7 \pm 0.9$ | $47.3 \pm 0.9$ | $\mathbf{50.0 \pm 0.9}$ | $43.6 \pm 0.9$ |
|  | Instruct | FT$\leq$30 | $52.0 \pm 0.9$ | $56.5 \pm 0.8$ | $56.0 \pm 0.8$ | $59.3 \pm 0.5$ | $\mathbf{60.5 \pm 0.5}$ |
| CLIP ViT-L/14 | Qwen3-8B | ICL | $55.0 \pm 0.9$ | $48.5 \pm 0.9$ | $\mathbf{63.5 \pm 0.7}$ | $57.6 \pm 0.5$ | $60.3 \pm 0.5$ |
|  |  | FT$\leq$30 | $72.3 \pm 0.9$ | $69.1 \pm 0.7$ | $71.4 \pm 0.9$ | $80.4 \pm 0.6$ | $\mathbf{83.5 \pm 0.6}$ |

Table 3: Comparison of prompt distillation approaches under different LMM settings while keeping the attention-mapper and soft prompts fixed. We report mean accuracy across 1 to 5 shots with 95% binomial confidence intervals for the OPEN_MI benchmark. The original LLaVA-ATT-Qwen2.5 7B architecture is highlighted in gray. FT: Finetuning with $K \leq 30$ gradient steps, ICL: In-Context Learning, TTA: Test-Time Adaptation. NoMeta-task$^{PD}$ and Model-Avg$^{PD}$ do not use meta-tasks.

use of a fixed number of soft prompts is superior to In-Context$^{PD}$, where the number of soft prompts vary according to the number of shots, .

**To what extent does MAPD facilitate task understanding at test time?** We take a closer look at how effectively MAPD captures task understanding at time, using the Operator Induction task (See Figure 8) as a case study. To solve this task, the model should correctly (a) identify the operands in the query example (*Perception*); (b) identify the operation from few-shot examples (*Task Induction*); and (c) use its own mathematical knowledge over the identified elements to reason towards the answer (*Mathematical Reasoning*). To test whether the model understands these subtasks, we design specific prompts and modify query examples as listed in Appendix A.2.9. In Figure 4(b), we observe that MAPD outperforms other prompt distillation approaches on all three subtasks, leading to better performance on the original task. MAPD shows a major improvement for task induction with an increase of 11.7% compared to MultiTask$^{PD}$. We also observe that solving each subtask individually is easier than tackling the original task, as the latter requires integrating knowledge from all subtasks, which is challenging when only a few shots are available at test time. Finally, MAPD excels at mathematical reasoning, effectively utilizing the underlying LLM's reasoning capabilities.

**Does MAPD generalize across different LMM architectures?** We next examine different LMM architectures that affect MAPD's performance. Specifically, we report results in four settings that vary the underlying LLM and vision encoder: a) using a smaller LLM (Qwen2.5-3B-Instruct); b) using a different and relatively weaker LLM (Vicuna v1.5-7B); c) using a different vision encoder, SigLIP (Zhai et al., 2023); and d) and using a relatively powerful LLM (Qwen3-8B) (Yang et al., 2025). In Table 3, MAPD outperforms other baselines with FT adaptation across different model configurations on the Open_MI task, demonstrating its robustness and generalizability. Fine-tuning based test-time adaptation for prompt distillation substantially outperforms ICL, with average improvements of +24.6, +37.06, +12.02, and +18.04 across the four settings, respectively. This highlights the significant benefits of test-time prompt distillation.

## 5 CONCLUSION

This work introduced Meta-Adaptive Prompt Distillation (MAPD), a novel meta-learning approach that endows LMMs with few-shot capabilities. MAPD employs a fixed set of soft prompts, distilled from task-relevant image features, which can be adapted at test time using only a few examples. A key component of our method is an attention-mapper module, which can be flexibly integrated with any LMM architectures and is jointly learned with soft prompts to facilitate distillation. Extensive evaluation on the VL-ICL benchmark shows that MAPD consistently outperforms traditional ICL and other efficient finetuning approaches across a diverse range of VQA tasks. Additional analysis (Appendix A.3) shows that MAPD incurs higher per-query latency than ICL due to gradient-based adaptation, but scales more favorably with increased test-time compute budget and is more data-efficient. Future work could focus on improving MAPD's computational efficiency for resource-constrained scenarios and extending it to multi-image tasks and complex reasoning problems.

ACKNOWLEDEGMENTS

We thank the anonymous reviewers for their feedback. We gratefully acknowledge the support of the UK Engineering and Physical Sciences Research Council (grant EP/W002876/1).

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

# A APPENDIX

Table 4: Finetuning Data Mixture Statistics

| Dataset | No. of examples | Question Types |
|---|---|---|
| LLaVA-Instruct | 157,712 | Conversations (57,669)
Detailed Image Description (23,240)
Complex Reasoning (76,803) |
| GQA | 72,140 | Visual Reasoning |
| OCR-VQA | 80,000 | Image Question Answering
with Reading Comprehension |
| TextVQA | 21,953 | Image Question Answering
with Reading Comprehension |
| Visual Genome | 86,417 | Image Question Answering
and Bounding Box Prediction |
| MAVIS-Math-Metagen | 87,348 | Visual Math
Question Answering |
| TabMWP-Cauldron | 22,717 | Tabular Math Reasoning |
| RefCOCO | 48,447 | Image Question Answering
and Bounding Box Prediction |
| OKVQA | 8,998 | Knowledge Grounded
Image Question Answering |
| VQAv2 | 82,783 | Image Question Answering |
| A-OKVQA | 66,160 | Multiple-Choice Question
Answering |
| Geo-170k (QA) | 67,823 | Math Question Answering
and Reasoning |
| Total | 802,498 | |

## A.1 IMPLEMENTATION DETAILS

### A.1.1 FINETUNING DATA MIXTURE

For model finetuning, we create our multi-task data mixture for single image per example using the visual instruction tuning data of LLaVA v1.5 (Liu et al., 2023) which contains a mixture of 12 different datasets[6] ranging from long conversations to academic multiple-choice questions. Since we are only training image-based prompts, we remove the language-only ShareGPT-40K dataset (ShareGPT, 2023). Additionally, we include 3 different math reasoning/QA datasets from the LLaVA OneVision data mixture (Li et al., 2025) which are known to improve LMM performance on difficult reasoning and logical QA tasks (Lu et al., 2024). We further get rid of the extra answer formatting instructions to test the true few-shot transfer learning ability of our approach without the need of external task induction. Table 4 shows the list of all the datasets along with their size and question types.

### A.1.2 DETAILS ON META-TASK CREATION

As mentioned in Section 3.2, meta-tasks are small subsets of examples randomly sampled from a single VQA dataset ($D^i$) within the training data mixture ($p(\mathcal{D})$). Each meta-task consists of support and query sets, each containing a fixed number of VQA examples (image, question, answer triplets). The support set provides few-shot demonstrations to the model, either as in-context examples or for gradient-based adaptation, depending on the prompt distillation method. The query set is used to

---

[6]We use this dataset only for academic research purposes as mentioned by the original authors and follow the Open AI Usage Policy for GPT-4 generated datasets. Additionally, we conform to the license (CC-BY-4.0) for Cauldron datasets.

Table 5: Meta-Task composition during test-time adaptation

| | Open_MI | Operator Induction | CLEVR | TextOCR |
|---|---|---|---|---|
| No. of Meta-tasks | 5000 | 4000 | 6000 | 5000 |
| Support examples per meta-task | [1,2,4,5] | [1,2,4,8] | [1,2,4,8] | [1,2,4,8] |
| Query examples per meta-task | 1 | 1 | 1 | 1 |

Table 6: Meta-Task composition during the finetuning stage

| | MAPD | Multi-task[PD] | In-Context[PD] |
|---|---|---|---|
| No. of Meta-tasks (train/val) | 39,650 / 8000 | 79,300 / 8000 | 72,100 / 8000 |
| Support examples per meta-task (train/val) | 10 / [1,2,4,5,8] | 5 / [1,2,4,5,8] | 10 / [1,2,4,5,8] |
| Query examples per meta-task (train/val) | 10 / 1 | 5 / 1 | 1 / 1 |
| Total no. of examples (train/val) | 793,000 / 16,000 | 793,000 / 16,000 | 793,000 / 16,000 |

optimize the LMM (specifically, the attention-mapper parameters ($\theta_p$) in our case) during fine-tuning, and to evaluate performance during inference. This meta-task construction protocol remains consistent across both the fine-tuning stage and test-time fine-tuning, following the framework established by (Zong et al., 2025).

During test-time adaptation, we use the publicly available VL-ICL benchmark code[7] to construct meta-tasks of fixed sizes. VQA examples are randomly sampled from the predefined training and test splits of each dataset. Table 5 specifies the number of meta-tasks per test set, which remains constant throughout our evaluation. All results reported in the paper represent average accuracy computed over the query examples of these meta-tasks, ensuring fair comparison across all prompt distillation methods and shot configurations.

During the attention-mapper fine-tuning stage, in order to keep a balanced ratio of train-validation splits across multiple datasets in Section A.1.1 used in this stage, we divide each dataset into 98% for training and 2% for validation and then combine them separately to create the final train and validation splits. We then construct meta-tasks by randomly sampling VQA examples. We treat the support-query composition as a tunable hyperparameter alongside those listed in Table, performing a grid search to identify the configuration that minimizes validation loss for each prompt distillation method. Table 6 details the optimal support-query compositions, number of meta-tasks, and total number of training and validation examples used for each method.

Additionally, for In-Context[PD], we follow the in-context tuning algorithm of (Chen et al., 2022), which uses only 1 query example per meta-task during training and yields optimal performance for this prompt distillation method. Note that the validation is done across a different number of support examples for robustness and the total number of training and validation examples remains constant across all methods to ensure fair comparison, regardless of meta-task composition.

---

[7]VL-ICL: https://github.com/ys-zong/VL-ICL.

Table 7: Grid search values for meta-task methods

|  | No. of support/query per meta task | Learning Rate | Inner-loop learning rate | Batch Size (# of meta-tasks) |
|---|---|---|---|---|
| Search Values | [1, 5, 10, 15] | [1e-3, 5e-4, 2e-5] | [1e-1, 5e-2, 5e-1] | [1, 5] |

Table 8: Grid search values for non meta-task methods

|  | Learning Rate | Batch Size |
|---|---|---|
| Search Values | [1e-3, 5e-4, 2e-5] | [16, 32, 64, 80] |

### A.1.3  MODEL CONFIGURATIONS

**Models** We use the publicly available implementation of LLaVA v1.5[8] and first-order MAML[9] to implement our baselines. Additionally, we use the pretrained model weights from Huggingface for Qwen2.5-7B-Instruct LLM[10] and the CLIP ViT-L/14-336px visual encoder[11]. The output embedding dimension size of CLIP is 1,024 and the input word embedding size of the Qwen LLM is 3,584. We set the training context length as 4096 for all baselines except for in-context baseline where it is 8,192 as it requires training with longer sequences. The attention-mapper is a single multi-head attention block with 8 heads. The token length of the soft prompt $P$ as described in Section 3.3 for the attention mapper is set to $m = 256$. The total number of trainable parameters for our model is approximately 24M making our approach significantly parameter-efficient for finetuning.

### A.1.4  TRAINING DETAILS

**Pretraining Stage** During the pretraining stage, we only train the attention-mapper and soft prompts for 4 epochs and use a learning rate of 2e-3 with a batch size of 64 per GPU. We perform a train-validation split on the LCS-558K dataset (Liu et al., 2023) by keeping 98% of the examples for training and 2% for validation and take the checkpoint with the lowest validation loss. We use this checkpoint as our base for further task-specific finetuning.

**Finetuning Stage** For finetuning, we perform a grid search across a fixed set of values as we are constrained by our GPU resources (4 H200 GPUs). For each prompt distillation method, we select the configuration that achieves the lowest validation loss following standard train-val-test procedures. Table 7 (for meta-task methods) and Table 8 (for non-meta-task methods) provide details of all hyperparameters for which we performed grid search. We also provide additional training details below, separately for each method, along with their corresponding best sets of hyperparameters after grid search. All approaches were finetuned for 1 epoch to ensure a complete pass over the entire finetuning data mixture.

1. **MAPD:** We use 5 inner-loop steps and initialize the inner-loop learning rate $\alpha$=1e-1. The outer-loop learning rate is set as 1e-3 with a per GPU batch size of 1 meta-task with a gradient accumulation of 2 steps. Each meta-task for training contains 10 support and 10 query examples. Training time $\sim$ 10 hours.

2. **Multi-Task[PD]:** Similar to MAPD, we use a learning rate of 1e-3 with a per GPU batch size of 1 meta-task with a gradient accumulation of 4 steps. Each meta-task for training contains 5 support and 5 query examples. Training time $\sim$ 4.5 hours

3. **In-Context[PD]:** We use a learning rate of 1e-3 with a gradient accumulation of 4 steps and 5 meta tasks per GPU. Each meta task for training contains 10 support examples and 1 query example. The support examples were concatenated with the strategy that ensured all image tokens of a meta-task are present in the sequence and we truncate the text tokens

---

[8]LLaVA v1.5: `https://github.com/haotian-liu/LLaVA/tree/main/llava`

[9]MAML: `https://github.com/AntreasAntoniou/HowToTrainYourMAMLPytorch`

[10]Qwen2.5-7B-Instruct: `https://huggingface.co/Qwen/Qwen2.5-7B-Instruct`

[11]CLIP-ViT-L/14-336px: `https://huggingface.co/openai/clip-vit-large-patch14-336`

if the sequence exceeded the context length of 8192. Further, the few-shot question and answers were concatenated by inserting "Question:" and "Answer:" strings in between them, inspired from (Alayrac et al., 2022). Training time $\sim 4.5$ hours

4. **ModelAvg$^{\text{PD}}$:** We first finetune individual models on each dataset in the finetuning data mixture (Section A.1.1) with a learning rate of 5e-4. For all the datasets, we choose a per GPU batch size of 8 with gradient accumulation of 2 steps. Average time per dataset $\sim 3$ hours

5. **NoMeta-task$^{\text{PD}}$:** Here, we finetune on the complete data mixture in one training run by sampling batches randomly and again use a per GPU batch size of 8 with a gradient accumulation of 2 steps. We also use a learning rate of 5e-4. Training time $\sim 4$ hours.

6. **LoRA**: We only apply LoRA to the attention matrices $(Q, K, V)$ of each layer. For training, we use a learning rate of 5e-4 and a per GPU batch size of 8 with gradient accumulation of 2 steps. Further, we performed hyperparameter search for choosing LoRA parameters - rank (r) and scaling factor ($\alpha$) for the three settings shown in Table 2. Training time $\sim 4$ hours.

   (a) *All LLM layers* ($r = 128, \alpha = 256$)

   (b) *[0-15] LLM layers* ($r = 16, \alpha = 64$)

   (c) *[0-15] LLM layers + ATT*: ($r = 16, \alpha = 64$)

**Computational Requirements** We find that the GPU requirement for training the attention-mapper mostly depends on the size of the underlying LLM used. For the 7B model training, we use 4 H200 GPUs with a VRAM of 143GB per GPU and for 3B models only 2 H200 GPUs were needed. For both the stages, the hyperparameters were tuned using their corresponding validation sets and we choose the checkpoints at the end of first epoch to report our results.

### A.1.5 PSEUDO ALGORITHM FOR MAPD

We highlight our full MAPD algorithm based on FoMAML in detail with inner and outer loop that is used to train the attention-mapper parameters $\theta_p$ in Algorithm 1.

---

**Algorithm 1:** Meta-Adaptive Prompt Distillation (MAPD)

---

**Input:** Meta-Task distribution $p(\mathcal{T}^{\text{meta}})$, inner-loop learning rate $\alpha$, meta learning rate $\beta$
**Output:** Meta-parameters $\theta_p = \{\theta, P\}$
Initialize $\theta_p$ with Xavier Uniform Initialization;
**while** *not converged* **do**

    Sample batch of meta-tasks $\{T_j\}_{j=1}^N \sim p(\mathcal{T}^{\text{meta}})$;

    **foreach** *task* $T_j = \{D_j^{supp}, D_j^{query}\}$ *in batch* **do**

        Evaluate $L_{\theta_{p,j}}^{\text{supp}} = \dfrac{-1}{|D_j^{\text{supp}}|} \sum_{i=1}^{|D_j^{\text{supp}}|} \log(p_{\theta_{p,j}}(X_a^i | X_v^i, X_q^i))$;

        Adapt parameters with $K$ gradient steps:

        **for** $k = 1, \ldots, K$ **do**

$$\theta_{p,j}^k \leftarrow \theta_{p,j}^{k-1} - \alpha \nabla_{\theta_{p,j}^{k-1}} L_{\theta_{p,j}^{k-1}}^{\text{supp}}$$

    Evaluate $L_{\theta_{p,j}^K}^{\text{query}} = \dfrac{-1}{|D_j^{\text{query}}|} \sum_{i=1}^{|D_j^{\text{query}}|} \log(p_{\theta_{p,j}^K}(X_a^i | X_v^i, X_q^i))$;

    First-Order Meta-Update:

$$\theta_p \leftarrow \theta_p - \beta \sum_{j=1}^N \nabla_{\theta_{p,j}^K} L_{\theta_{p,j}^K}^{\text{query}}$$

---

- **Operator Induction** - *"The image contains two digit numbers and a ? representing the mathematical operator. Induce the mathematical operator (addition, multiplication, minus) according to the results of the in-context examples and calculate the result."*
- **CLEVR Count Induction** - *"The image contains objects of different shapes, colors, sizes and materials. The question describes the attribute and its value. You need to find all objects within the image that satisfy the condition. You should induce what operation to use according to the results of the in-context examples and then calculate the result."*

Figure 6: Detailed task instruction for LLaVA-OneVision-7B LMM evaluation on VL-ICL tasks.

Table 9: Evaluation Dataset Statistics

| Dataset | Task Category | Train Set (Support) | Test Set (Query) | Size (GB) |
|---|---|---|---|---|
| Fast Open-MiniImageNet (OPEN_MI) | Fast-Concept Binding | 5,000 | 200 | 0.18 |
| CLEVR Count Induction | Fine-Grained Perception, Task Induction | 800 | 200 | 0.18 |
| Operator Induction | Perception, Task Induction Mathematical Reasoning | 80 | 60 | 0.01 |
| TextOCR | Perception, Task Induction | 800 | 200 | 0.01 |

## A.2 EVALUATION DETAILS

### A.2.1 DETAILED TASK INSTRUCTIONS FOR LMM EVALUATION.

Figure 6 shows the detailed task instructions used to evaluate LLaVA-OneVision-7B in the Image-to-Text (I2T) ICL setting.

### A.2.2 EVALUATION DATASETS FROM VL-ICL BENCH

The VL-ICL Bench Zong et al. (2025) includes a diverse variety of tasks to test different capabilities of models like Fast-Concept binding, Mathematical Induction, and Fine-grained perception. Given the nature of our model architecture and training (Section 3), we only focus on the single-image Image-to-text (I2T) tasks. Table 9 shows the dataset statistics. We also give brief descriptions of these tasks below along with some examples for better understanding.

1. **Fast Open-Ended MiniImageNet (OPEN_MI)** - This is a variant of the MiniImageNet few-shot object recognition task (Vinyals et al., 2016), which was repurposed for few-shot prompting (Tsimpoukelli et al., 2021). It is essentially an open-ended image classfication problem, but contains nonsense categorical names like *dax* or *blicket* making the test performance not influenced by the prior knowledge of an LMM but only dependent on the support examples. This design ensures to test the few-shot abilities of LMMs and how quickly they can learn about new concepts. For the results shown in Table 11, we use the 2-way version of this task involving classification between two nonsense categories. An example of a 2-way 1-shot task is shown in Figure 7.

2. **Operator Induction** - Initially proposed by (Zong et al., 2025), this dataset tests various capabilties of LMMs like Task Induction, Perception and Mathematical Reasoning. The support examples involve two operands with a missing mathematical operation and an answer. When testing, the task is to identify the hidden operation from the support example and use it to calculate the result over the operands in the query. An example of a 2-shot task is shown in Figure 8.

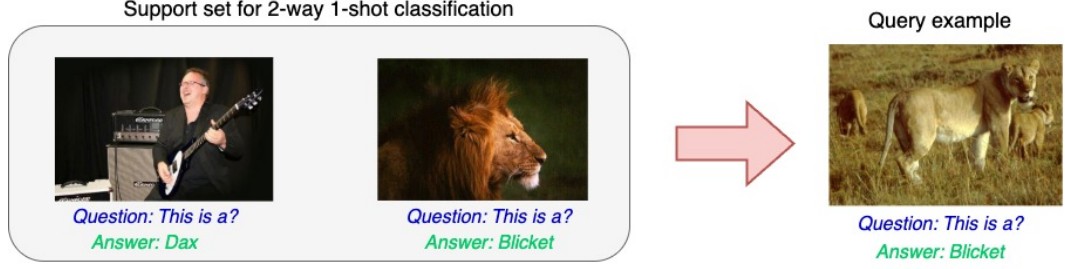

Figure 7: 2-way Fast Open-Ended MiniImageNet

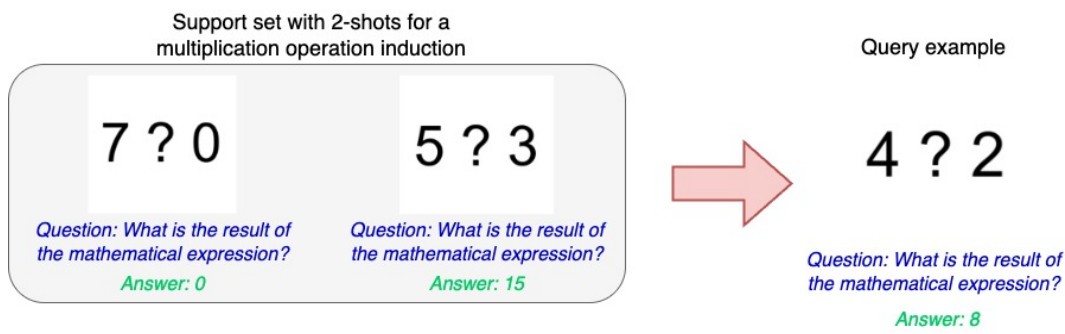

Figure 8: Operator Induction

3. **CLEVR Count Induction** - This dataset contains images from the widely used CLEVR dataset (Johnson et al., 2017) where each image contains a set of objects that have certain characteristics based on attributes like shape, size, color and material. The task is to learn to count the objects of the given attribute in the support example and transfer that knowledge to count the objects of any attribute in the query example. An example of a 2-shot task is shown in Figure 9.

4. **TextOCR** - This dataset has been repurposed by (Zong et al., 2025) from the TextOCR dataset (Singh et al., 2021) to create a task where the LMM should learn to output the text within a red bounding box from the support examples. Even though this task could be solved in a zero-shot setting as we see in the 0-shot case with a detailed prompt, we still only focus on inducing task knowledge from the few-shot examples. An example of a 2-shot task is shown in Figure 10.

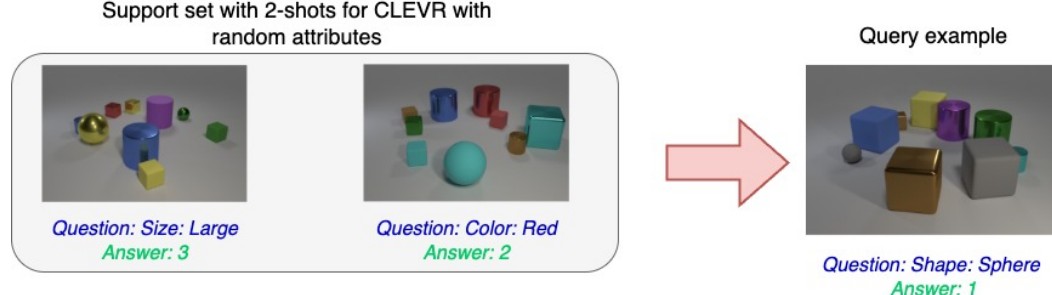

Figure 9: CLEVR Count Induction

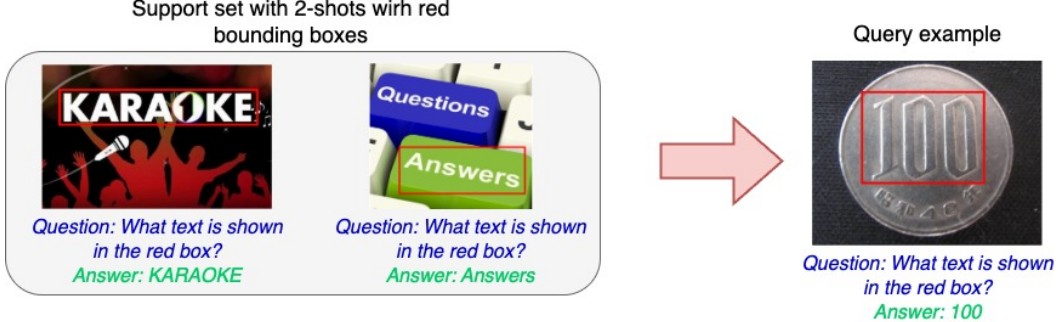

Figure 10: TextOCR

### A.2.3 TEST-TIME ADAPTATION DETAILS

We follow a similar test-time adaptation procedure to (Qin et al., 2023) to find the best hyperparameter settings for each prompt distillation method, ensuring fair comparison. We first sample 10% of the examples from the training split of each test task and combine them to form a validation set. After meta-task creation over the VL-ICL datasets (Zong et al., 2025) using the remaining training and test splits, we perform a maximum of $K = 30$ inner-loop steps over each support set and select the $K$th-step model that achieves the lowest validation loss. This model is then used to compute results on the query set. To further validate whether $K = 30$ is a sufficient number of fine-tuning steps, we plot average test accuracy curves (up to 40 gradient steps) across VL-ICL datasets, methods, and shot settings in Figure 11. Accuracies converge within 30 steps, confirming our choice of $K$. We also provide examples of how predictions evolve during test-time adaptation in Figures 12, 13, 14, and 15. For reproducibility, Table 10 reports the best learning rates for each method, selected via grid search over $[0.1, 1.0]$ on the validation set with a batch size of one meta-task.

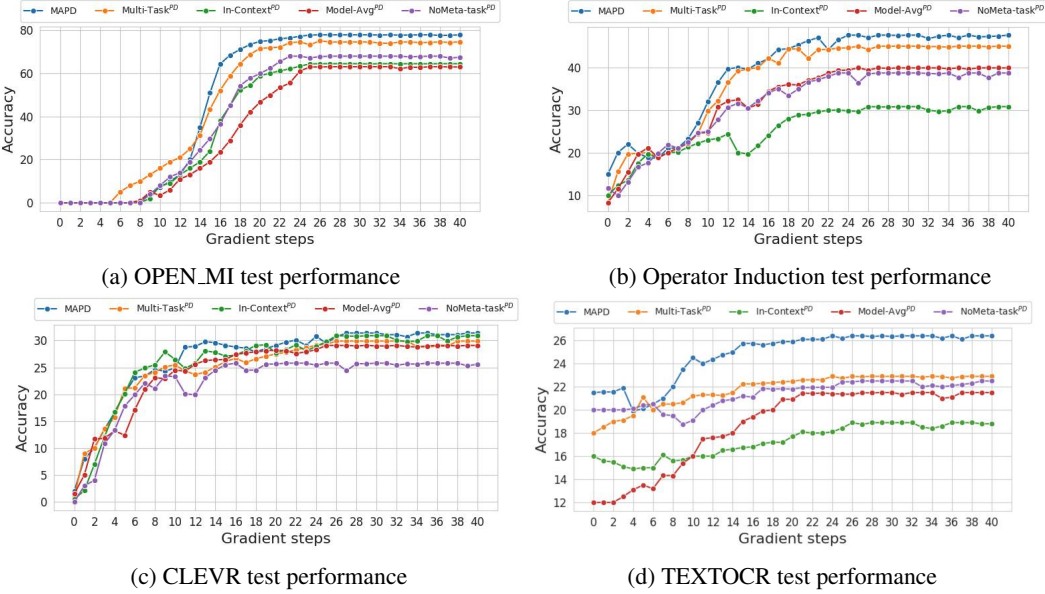

(a) OPEN_MI test performance  (b) Operator Induction test performance

(c) CLEVR test performance  (d) TEXTOCR test performance

Figure 11: Average test performances of MAPD with finetuning on different datasets

Table 10: Learning rates for finetuning-based (FT) test-time adaptation for results shown in Table 1, Table 2, Table 11 and Table 12.

| Training Methods | Learning Rate (LR) |
|---|---|
| MAPD | 1.0 |
| Multi-Task[PD] | 0.8 |
| In-Context[PD] | 0.8 |
| ModelAvg[PD] | 0.6 |
| NoMeta-task[PD] | 1.0 |
| LoRA | 0.2 |

### A.2.4 Detailed Results

Table 11: Comparison of different prompt distillation approaches on single-image tasks from VL-ICL Bench (Zong et al., 2025). We report accuracy for different numbers of shots (–S). "Avg" is only calculated for ≥ 1 shot(s). FT = Finetuning with ≤ 30 gradient steps, ICL = In-Context Learning, TTA= Test-Time Adaptation. More details are mentioned in Appendix A.2.3. We do not compare on 0-shot results. The model used for this evaluation is LLaVA-ATT-Qwen2.5 7B which is described in Section 3.3. Meta-Tasks used (✓) or not (✗) during training. We also provide results for higher number of shots in Appendix A.2.11 and qualitative results in Appendix A.2.6 and A.2.9.

| Methods | Meta Task | Open-MI (2-way) | | | | | | Operator Induction | | | | | |
| --- | --- | --- | --- | --- | --- | --- | --- | --- | --- | --- | --- | --- | --- |
| | | 0-S | 1–S | 2–S | 4–S | 5–S | Avg | 0-S | 1–S | 2–S | 4–S | 8–S | Avg |
| **TTA with ICL** | | | | | | | | | | | | | |
| NoMeta-task[PD] | ✗ | 0.0 | 35.0 | 47.0 | 48.0 | 45.0 | 43.8 | 11.7 | 13.3 | 13.3 | 10.0 | 11.7 | 12.1 |
| Model-Avg[PD] | ✗ | 0.0 | 20.0 | 22.0 | 30.0 | 34.5 | 26.6 | 8.3 | 11.7 | 6.7 | 8.3 | 10.0 | 9.2 |
| In-Context[PD] | ✓ | 0.0 | 30.0 | 56.0 | 55.0 | 63.5 | 51.1 | 10.0 | 20.0 | 18.5 | 18.0 | 26.0 | 20.6 |
| Multi-Task[PD] | ✓ | 0.0 | 43.0 | 50.0 | 51.0 | 50.5 | 48.6 | 8.3 | 13.3 | 11.7 | 3.3 | 11.7 | 10.0 |
| MAPD | ✓ | 0.0 | 42.5 | 53.0 | 57.0 | 60.5 | 53.3 | 15.0 | 13.3 | 13.3 | 1.7 | 10.0 | 9.6 |
| **TTA with FT ≤30** | | | | | | | | | | | | | |
| NoMeta-task[PD] | ✗ | 0.0 | 21.5 | 67.5 | 89.0 | 94.0 | 68.0 | 11.7 | 26.7 | 23.3 | 46.7 | 58.3 | 38.8 |
| Model-Avg[PD] | ✗ | 0.0 | 28.5 | 53.5 | 83.0 | 87.5 | 63.1 | 8.3 | 31.5 | 28.0 | 45.0 | 55.5 | 40.0 |
| In-Context[PD] | ✓ | 0.0 | 35.5 | 54.5 | 79.5 | 88.5 | 64.5 | 10.0 | 21.7 | 18.3 | 41.7 | 41.7 | 30.9 |
| Multi-Task[PD] | ✓ | 0.0 | 37.0 | 73.5 | 93.5 | 94.5 | 74.6 | 8.3 | 31.0 | 28.3 | **61.0** | 60.0 | 45.1 |
| **MAPD** | ✓ | 0.0 | **43.5** | **78.0** | **94.5** | **95.5** | **77.9** | 15.0 | **32.0** | **38.3** | 58.3 | **62.0** | **47.7** |

| Methods | Meta Task | CLEVR Count Induction | | | | | | TextOCR | | | | | |
| --- | --- | --- | --- | --- | --- | --- | --- | --- | --- | --- | --- | --- | --- |
| | | 0–S | 1–S | 2–S | 4–S | 8-S | Avg | 0–S | 1–S | 2–S | 4–S | 8–S | Avg |
| **TTA with ICL** | | | | | | | | | | | | | |
| NoMeta-task[PD] | ✗ | 0.0 | 8.0 | 10.5 | 23.0 | 30.5 | 18.0 | 20.0 | 4.5 | 9.5 | 8.5 | 4.5 | 6.8 |
| Model-Avg[PD] | ✗ | 1.5 | 17.0 | 8.5 | 4.0 | 1.0 | 7.6 | 12.0 | 3.0 | 2.5 | 3.0 | 1.0 | 2.8 |
| In-Context[PD] | ✓ | 0.0 | 13.5 | 23.0 | 28.5 | 31.5 | 24.1 | 16.0 | 22.5 | 21.0 | 23.5 | 28.0 | 23.8 |
| Multi-Task[PD] | ✓ | 1.0 | 5.0 | 9.0 | 16.5 | 19.5 | 12.5 | 18.0 | 4.0 | 4.5 | 8.5 | 10.5 | 6.9 |
| MAPD | ✓ | 2.0 | 11.0 | 7.0 | 15.5 | 15.5 | 12.3 | 21.5 | 5.5 | 7.0 | 8.0 | 8.5 | 7.3 |
| **TTA with FT ≤30** | | | | | | | | | | | | | |
| NoMeta-task[PD] | ✗ | 0.0 | 18.5 | 21.5 | 26.0 | 37.0 | 25.8 | 20.0 | 20.5 | 23.0 | 24.0 | 22.5 | 22.5 |
| Model-Avg[PD] | ✗ | 1.5 | **26.5** | 25.0 | 29.5 | 35.5 | 29.1 | 12.0 | 17.5 | 20.0 | 23.0 | 25.5 | 21.5 |
| In-Context[PD] | ✓ | 0.5 | 24.5 | **30** | **34.5** | 34.5 | 30.9 | 16.0 | 16.0 | 18.0 | 19.5 | 22.0 | 18.9 |
| Multi-Task[PD] | ✓ | 0.0 | 25.0 | 25.5 | 31.0 | 38.0 | 29.9 | 18.0 | 21.0 | 20.5 | 24.5 | 25.5 | 22.9 |
| **MAPD** | ✓ | 0.0 | **26.5** | 27.5 | 31.0 | **40.5** | **31.4** | 21.5 | **23.5** | **26.5** | **27.0** | **28.5** | **26.4** |

Table 12: Comparison of the LoRA baselines on VL-ICL Bench (Zong et al., 2025). "Avg" is only calculated for ≥ 1 shot(s) (-S). TTA= Test-Time Adaptation. FT=Finetuning with ≤ 30 gradient steps. ATT=Attention-Mapper. The model used for this evaluation is LLaVA-ATT-Qwen2.5 7B.

| LoRA | Open-MI (2-way) | | | | | | Operator Induction | | | | | |
| --- | --- | --- | --- | --- | --- | --- | --- | --- | --- | --- | --- | --- |
| | 0-S | 1–S | 2–S | 4–S | 5–S | Avg | 0-S | 1–S | 2–S | 4–S | 8–S | Avg |
| **TTA with FT ≤ 30** | | | | | | | | | | | | |
| All LLM layers | 0.0 | 24.5 | 45.7 | 68.3 | 81.9 | 55.1 | 8.1 | 11.7 | 10.0 | 13.3 | 18.2 | 13.3 |
| [0-15] LLM layers | 0.0 | 30.9 | **65.3** | 81.1 | **91.9** | 67.3 | 8.3 | 18.3 | 26.3 | 23.1 | 34.3 | 25.5 |
| [0-15] LLM layers + ATT | 0.0 | **37.3** | 64.1 | **83.5** | 91.5 | **69.1** | 10.0 | **21.5** | **28.3** | **35.5** | **36.7** | **30.5** |

| LoRA | CLEVR Count Induction | | | | | | TextOCR | | | | | |
| --- | --- | --- | --- | --- | --- | --- | --- | --- | --- | --- | --- | --- |
| | 0-S | 1–S | 2–S | 4–S | 5-S | Avg | 0-S | 1–S | 2–S | 4–S | 8–S | Avg |
| **TTA with FT ≤ 30** | | | | | | | | | | | | |
| All LLM layers | 0.0 | 9.3 | 11.7 | 15.5 | 23.9 | 15.1 | 15.0 | 6.7 | 9.1 | 13.3 | 12.5 | 10.4 |
| [0-15] LLM layers | 0.0 | 21.5 | **28.3** | **32.5** | **37.7** | **30.0** | 18.3 | 20.3 | **24.5** | 25.5 | 24.9 | 23.8 |
| [0-15] LLM layers + ATT | 0.0 | **26.0** | 23.1 | 30.0 | 35.7 | 28.7 | 18.3 | **20.6** | 23.4 | **26.5** | **27.5** | **24.5** |

A.2.5 PERFORMANCE OF PUBLICLY AVAILABLE LMMS ON VL-ICL BENCH

Table 13: Performance of different LMMs on single-image tasks from VL-ICL Bench. We report the "Avg" accuracy for different numbers of shots - $\{1, 2, 4, 5, 8\}$ with 95% binomial confidence intervals. FT = Finetuning with $\leq 30$ gradient steps, ICL = In-Context Learning, TTA= Test-Time Adaptation, VL-Data=Vision-Language Data, LAQ-7B=LLaVA-ATT-Qwen2.5-7B, CLIP=CLIP-ViT-L/14-336px, MLP=2-layer MLP, ATT=Attention-Mapper. **Bold** shows best performance and Underline is MAPD's performance with LAQ-7B LMM.

| Methods | VL-Data | Params trained | TTA | Open-MI | OP_IND | CLEVR | TextOCR |
|---|---|---|---|---|---|---|---|
| LLaVA v1.5-7B | 1.2M | 7B | ICL | $12.4 \pm 0.4$ | $5.4 \pm 0.5$ | $10.9 \pm 0.1$ | $4.4 \pm 0.3$ |
| LLaVA v1.5-7B | 1.2M | 7B | FT$\leq$30 | $38.4 \pm 0.7$ | $11.4 \pm 0.6$ | $16.9 \pm 0.2$ | $15.6 \pm 0.6$ |
| LLaVA-Next-7B | 1.3M | 7.06B | ICL | $34.4 \pm 0.7$ | $5.4 \pm 0.5$ | $21.1 \pm 0.2$ | $0.4 \pm 0.0$ |
| LLaVA-Next-7B | 1.3M | 7.06B | FT$\leq$30 | $55.1 \pm 0.9$ | $13.4 \pm 0.6$ | $28.6 \pm 0.2$ | $7.8 \pm 0.4$ |
| LLaVA-OneVision-7B | 10.4M | 8B | ICL | $42.1 \pm 0.9$ | $41.7 \pm 0.5$ | $34.9 \pm 0.2$ | $42.3 \pm 0.5$ |
| LLaVA-OneVision-7B | 10.4M | 8B | FT$\leq$30 | $83.4 \pm 0.7$ | $46.1 \pm 0.5$ | $\mathbf{38.9 \pm 0.2}$ | $45.5 \pm 0.5$ |
| LLaVA-OneVision-72B | 10.4M | 73.2B | ICL | $75.1 \pm 0.6$ | $69.1 \pm 0.9$ | $37.2 \pm 0.2$ | $\mathbf{52.2 \pm 1.1}$ |
| Qwen2-VL-7B-Instruct | -NA- | 8B | ICL | $73.5 \pm 0.6$ | $69.6 \pm 0.9$ | $27.9 \pm 0.2$ | $50.5 \pm 0.9$ |
| Qwen2.5-VL-7B-Instruct | -NA- | 8B | ICL | $44.0 \pm 0.9$ | $84.2 \pm 1.2$ | $22.0 \pm 0.2$ | $36.9 \pm 0.7$ |
| Qwen2.5-VL-7B-Instruct | -NA- | 8B | FT$\leq$30 | $\mathbf{85.6 \pm 0.7}$ | $\mathbf{89.4 \pm 1.2}$ | $29.1 \pm 0.2$ | $41.1 \pm 0.5$ |
| LAQ-7B + In-Context[PD] | 1.3M | 24M | ICL | $51.1 \pm 0.9$ | $20.6 \pm 0.8$ | $24.1 \pm 0.2$ | $23.8 \pm 0.8$ |
| LAQ-7B + In-Context[PD] | 1.3M | 24M | FT$\leq$30 | $64.5 \pm 0.8$ | $30.9 \pm 0.5$ | $30.9 \pm 0.2$ | $18.9 \pm 0.7$ |
| LAQ-7B + MAPD | 1.3M | 24M | ICL | $53.3 \pm 0.9$ | $9.6 \pm 0.5$ | $12.3 \pm 0.1$ | $7.3 \pm 0.4$ |
| **LAQ-7B + MAPD** | 1.3M | 24M | FT$\leq$30 | $77.9 \pm 0.7$ | $47.7 \pm 0.5$ | $31.4 \pm 0.2$ | $26.4 \pm 0.8$ |

We report the performance of publicly available LMMs alongside our best-performing architecture (LLaVA-ATT-Qwen2.5-7B) on the single-image tasks from VL-ICL Bench in Table 13. **We provide this as a reference and note that direct comparison across LMMs is not straightforward, given their fundamental differences in architecture, scale, and training data.**

1. Test-time fine-tuning of the MLP connector consistently improves over ICL for all public LMMs, supporting our hypothesis that these models are overwhelmed by image embeddings during ICL. Fine-tuning enables the connector to distil task-specific information into image embeddings before prompting the LLM, thereby improving few-shot performance.

2. Our model with MAPD-based meta-learning and fine-tuning adaptation performs comparably to other publicly available LMMs and, notably, surpasses LLaVA-OneVision-72B ICL on the Fast Open-Ended MiniImageNet (Open-MI) task, as well as ICL with its 7B counterpart (trained on substantially more data) and the stronger Qwen-VL models on other tasks.

3. Unlike other LMMs, LLaVA-ATT-Qwen2.5-7B (LAQ-7B) does not fine-tune the LLM during training and uses significantly less vision-language data (1.3M examples) and fewer trainable parameters (24M), compared to LLaVA-OneVision which trains the full model on 10.4M examples. This highlights the data and parameter efficiency of MAPD, which achieves state-of-the-art performance on Open-MI by fine-tuning only the attention-mapper for up to 30 gradient steps on the few-shot examples.

4. For LMMs such as LLaVA-OneVision, fine-tuning the attention-mapper requires substantial compute ($\geq$12 H200 GPUs) due to their large-scale training mixture (10.4M vision-language examples) and high-dimensional vision encoder embeddings, exceeding our available resources. Similarly, fine-tuning data for the Qwen-VL models is not publicly available. Given these constraints, we are unable to conduct attention-mapper fine-tuning experiments on these architectures.

### A.2.6 QUALITATIVE RESULTS

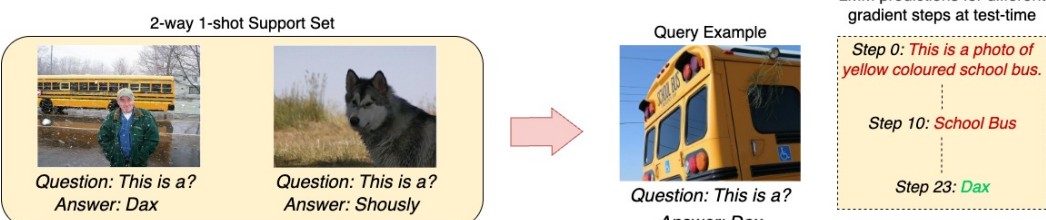

Figure 12: OPEN_MI predictions at test-time

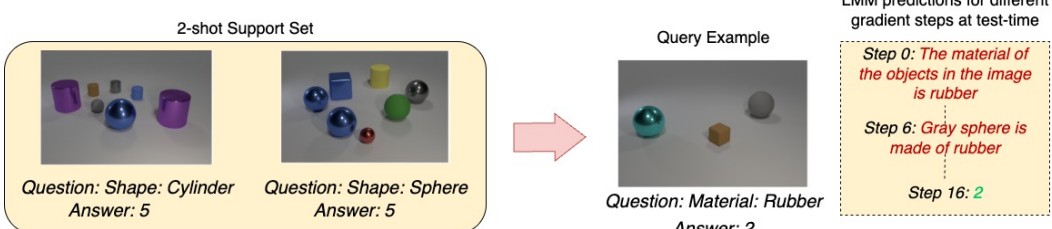

Figure 13: CLEVR predictions at test-time

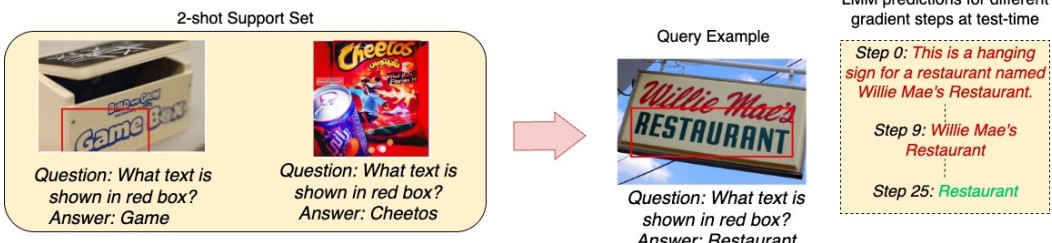

Figure 14: TEXTOCR predictions at test-time

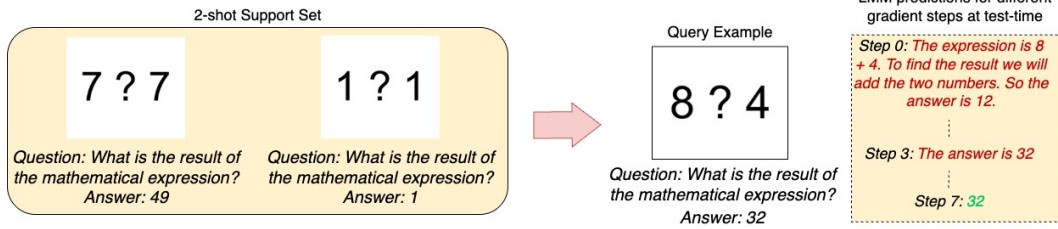

Figure 15: Operator Induction predictions at test-time

Table 14: Robustness of prompt distillation methods against image perturbations on the Fast Open-Ended MiniImageNet dataset (2-way classification) for LLaVA-ATT-Qwen2.5 7B LMM. We report accuracy scores as defined in VL-ICL Bench (Zong et al., 2025) across 2, and 5 shots. Test-Time Adaptation = Finetuning with ≤ 30 gradient steps.

| | NoMeta-task[PD] | | Model-Avg[PD] | | In-Context[PD] | | Multi-task[PD] | | MAPD | |
|---|---|---|---|---|---|---|---|---|---|---|
| | 2–S | 5–S | 2–S | 5–S | 2–S | 5-S | 2–S | 5–S | 2–S | 5–S |
| Original | 67.5 | 94.0 | 53.5 | 87.5 | 54.5 | 88.5 | 73.5 | 94.5 | 78.0 | 95.5 |
| Cropping | 65.0 | 94.0 | 51.5 | 87.5 | 51.5 | 83.0 | 72.0 | 91.5 | 76.5 | 95.0 |
| Rotation | 67.0 | 91.0 | 50.5 | 81.5 | 50.5 | 83.5 | 72.5 | 93.5 | 78.0 | 95.5 |
| Gaussian Blur | 67.5 | 92.5 | 51.5 | 84.5 | 49.5 | 78.0 | 71.5 | 92.5 | 77.5 | 96.0 |
| Color Jitter | 66.5 | 92.5 | 50.5 | 89.0 | 49.5 | 81.5 | 71.5 | 94.0 | 77.0 | 94.0 |
| CutMix | 58.5 | 86.0 | 45.5 | 70.5 | 49.0 | 75.0 | 72.0 | 92.0 | 75.5 | 92.5 |
| MixUp | 58.0 | 84.0 | 46.0 | 70.5 | 48.0 | 75.5 | 69.0 | 89.0 | 76.5 | 91.0 |
| Mean Drop in Accuracy | −3.8 | −4.0 | −4.3 | −6.9 | −4.8 | −9.1 | −2.1 | −2.4 | −1.2 | −1.4 |
| Net Mean Drop across Shots | −3.9 | | −5.6 | | −7.0 | | −2.3 | | **−1.3** | |

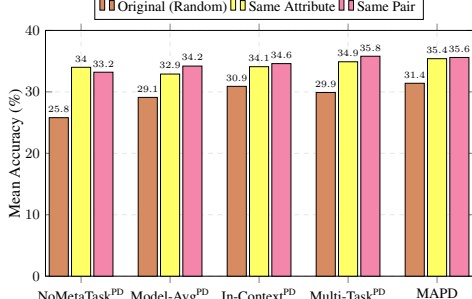
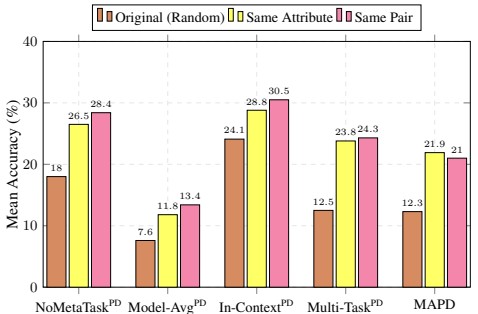

Figure 16: (a) Performance comparison of different prompt distillation approaches on the CLEVR Count Induction (details in Appendix A.2.2). Few-shot examples for *Same Attribute* and *Same Pair* are selected based on their *attribute-value* similarity with the query (test) example. Mean Accuracy is computed for 1,2,4 and 8 shots. **Left**: Finetuning (FT) based Test-time Adaptation. (b) **Right**: In-Context Learning (ICL) based Test-time Adaptation.

### A.2.7 ROBUSTNESS AGAINST IMAGE PERTURBATIONS

We assess if our prompt distillation methods are robust enough to handle perturbations applied to the images in the support set as shown in Table 14. We see that our method, MAPD, is most robust even in the presence of noise in the support examples as compared to other distillation methods that suffer a huge drop in performance. Advanced techniques like CutMix (Yun et al., 2019) and MixUp (Zhang et al., 2018) change the original image distribution substantially, affecting all methods to a greater degree but MAPD is still close to its original performance for both 2 and 5 shots. This robustness likely stems from MAPD's meta-learned initialization, which learns the underlying task structure from meta-tasks without over-fitting to any other spurious visual patterns and this allows it to adapt quickly to newer tasks without being influenced by noisy visual artifacts in the examples.

### A.2.8 HOW TO SELECT FEW-SHOT EXAMPLES FOR BETTER PERFORMANCE?

We further assess how performance varies for different prompt distillation approaches based on the selection of few-shot examples on the CLEVR Count Induction task (details in Appendix A.2.2) as an example. We propose two selection methods based on similarity of attributes and their corresponding values for every query (test) example. If the query has attribute and value as *shape: sphere*, we select the few-shot examples based on - a) Same Attribute - *shape*, (b) Same Pair - *shape: sphere* and compare both of them with the original setup as proposed in the VL-ICL benchmark (Zong et al., 2025) which retrieves the few-shot examples randomly. In Figure 16(a), we first see that for finetuning-based (FT) adaptation, the performance of all the baselines increases by 4.8% for Same Attribute and 5.3% for Same Pair on an average. MAPD performs best in the Same Attribute

setting (Mean Acc = 35.4%) and Multi-Task[PD] performing best in the Same Pair setting (Mean Acc = 35.8%). In Figure 16(b), we see that for In-Context Learning (ICL) adaptation, the similarity-based few-shot selection methods have a greater impact in performance and improve the mean accuracy of all the baselines by 7.7% for Same Attribute and 8.6% for Same Pair on an average. In-Context[PD] performs the best in both Same Attribute and Same Pair settings with mean accuracies of 28.8% and 30.5% respectively for ICL adaptation. We also notice that the Same Pair setup is generally the best few-shot selection method giving best performance for all the approaches. This indicates that choosing few-shot examples that are similar to query example induces better task understanding during test-time adaptation. We also see that the selection of few-shot examples shows less variance with FT adaptation compared to ICL adaptation, thereby showing higher robustness of FT adaptation.

### A.2.9 DETAILS ON ABLATION STUDY FOR OPERATOR INDUCTION

We break down the ablation study on operator induction tasks (Section 4.3; Figure 4(b)) into 3 components: 1) Task Induction, 2) Perception, and 3) Mathematical Reasoning. We test these components separately with the help of suitable prompts for our LMM to answer questions in specific formats. Figure 17 shows our prompts used for different components.

> - **Task Induction** - *"What mathematical operation should be used in this example? Strictly answer in one word."*
> - **Perception** - *"What are the numbers in this example? Do not calculate the answer after applying mathematical operation. Only give the numbers shown in the example. Stricly give numbers in numeric digits and your result should be in the format > Number A: xxx || Number B: xxx."*
> - **Mathematical Reasoning** - *"Think step-by-step and give proper reasoning steps first and then given your final answer. The format should be > Reasoning: xxx || Answer: xxx . The Reasoning part should contain reasons to derive the answer and the Answer part should only contain the answer. Your response should strictly follow this format and not just give the answer of the mathematical operation. It's important that you give reasoning before you answer."*

Figure 17: (Operator Induction Task) Prompts to the LMM for generating answers in specific formats suited for evaluation.

We list out a few examples which we curate for the Operator Induction task to enhance mathematical reasoning. Each image in the dataset contains a set of 2 numbers or operands and a hidden mathematical operation. The result of the correct mathematical operation is also provided for the support set examples. The task is to induce the mathematical operation used in the support set to calculate the answer of the query image containing two new operands. As finetuning on a single answer token limits the token generation capacity of the LMM, we further modify the support set examples to list out detailed mathematical steps before calculating the answer. Finetuning on this reasoning data improves both the generation capacity and reasoning ability of the LMM. We further provide a few examples of this hand-curated data in Figure 18.

We used Qwen2.5VL-32B-Instruct (Qwen et al., 2025) as a judge for evaluating the Mathematical Reasoning component of the problem where LMMs responded with detailed reasoning steps before the answer. Evaluation of responses was done by prompting the judge to score a response between 0–3 based on if it thinks the reasoning and answer are correct. We then calculated mean score as the percentage of total score assigned by the Qwen-2.5-VL (Judge) to the responses relative to the maximum possible score.

$$\text{Mean Percent Score} = \frac{\sum_{i=1}^{N} S_i}{3 * N} \times 100 \tag{9}$$

where $S_i$ is the score assigned by Qwen2.5-VL for the ith response and $N$ is the total number of responses. We provide the prompt to the judge for this evaluation in Figure 19.

We also provide a few examples of LMM predictions for task induction (Figure 20) and perception (Figure 21) and mathematical reasoning (Example 1: Figure 22, 23 and Example 2: Figure 24, 25)

**Original Answer**: *4*
**Detailed Answer**: *There are two numbers, $5$ and $1$. Performing some mathematical operation gives the answer $4$. So if we think about adding the numbers, $5 + 1 = 6$, subtracting them, $5 - 1 = 4$, multiplying them, $5 \times 1 = 5$. This implies that the hidden operation must be subtraction (–) and the result is $4$.*

**Original Answer**: *21*
**Detailed Answer**: *There are two numbers, $3$ and $7$. Performing some mathematical operation gives the answer $21$. So if we think about multiplying the numbers, $3 \times 7 = 21$, adding the numbers, $3 + 7 = 10$, subtracting the numbers, $3 - 7 = -4$. This implies that the hidden operation must be multiplication or $\times$ and the result is $21$.*

Figure 18: (Operator Induction Math Reasoning) Few examples of our hand-curated data with mathematical reasoning steps.

**Judge Prompt** - *"You are given a few in-context examples of a mathematical induction problem. The in-context examples each have an image with two numbers and a '?' which is supposed to be some mathematical operation. You are given a solution that gives the answer and the reasoning on how to calculate that answer using some mathematical operation applied on those two numbers in the image. The task is to induce the correct mathematical operation from the given examples, and use that operation to calculate the result of a query image with different numbers.*

*After this, you are then given a reference answer written by experts and a candidate response. The candidate response is in format Reasoning: xxx || Answer: xxx . The reasoning part contains reasoning about how the candidate arrived at the solution, and the Answer part contains their final answer. Your task is to judge if the reasoning and the answer of the candidate response are correct or not after considering the in-context examples, query image, question, reference answer, and your own reasoning of the mathematical induction problem.*

*The rating should be done on a scale of 0–3, where 0 indicates when the response is ambiguous or does not follow the format, 1 is for when both the reasoning and answer are wrong, 2 is for when either only reasoning or answer is correct, 3 is for when both the reasoning and answer are correct.*

*Be strict in your judgement and do not give a higher rating unless the candidate response contains accurate reasoning and exact answer. Thoroughly check each and every part of the candidate response and make sure it does not contain extra irrelevant operations or answers. If it does then give a lower rating accordingly. The candidate response should follow the format and conclude with the correct answer. If it does not, that means their answer is wrong. Also give your rationale before rating. Give the final rating as > Rating: xxx"*

Figure 19: (Operator Induction Math Reasoning) Prompts for the Qwen2.5VL-32B-Instruct to evaluate LMM responses on a scale of 0–3. It is given 1 to 4 in-context examples for understanding the mathematical induction task before the LMM (candidate) response for better evaluation.

Example (Task Induction): Support shot 1

7 ? 6

**Question**: *What is the result of the following mathematical expression?*
**Answer**: *42*

Example (Task Induction): Support shot 2

0 ? 8

**Question**: *What is the result of the following mathematical expression?*
**Answer**: *0*

Example (Task Induction): Query

3 ? 6

**Question**: *What mathematical operation should be used in this example? Strictly answer in one word.*
**LMM prediction**: *Multiplication*

Figure 20: (Operator Induction Task Induction) An example of a 2-shot task induction for multiplication operation

Example (Perception): Support shot 1

| | |
|---|---|
| **2 ? 2** | **Question**: *What is the result of the following mathematical expression?* 
 **Answer**: *0* |

Example (Perception): Support shot 2

| | |
|---|---|
| **2 ? 4** | **Question**: *What is the result of the following mathematical expression?* 
 **Answer**: *-2* |

Example (Perception): Query

| | |
|---|---|
| **5 ? 4** | **Question**: *What are the numbers in this example? Do not calculate the answer after applying mathematical operation. Only give the numbers shown in the example. Stricly give numbers in numeric digits and your result should be in the format > Number A: xxx || Number B: xxx.* 
 **LMM prediction**: *Number A: 5 || Number B: 4* |

Figure 21: (Operator Induction Perception) An example of a 2-shot perception task to detect operands

Example 1: Support

**Question**: *What is the result of the following mathematical expression?*
**Answer**: *There are two numbers, 1 and 8. Performing some mathematical operation gives the answer 8. So if we think about subtracting the numbers, $1 - 8 = -7$, multiplying the numbers, $1 \times 8 = 8$, adding the numbers, $1 + 8 = 9$. This implies that the hidden operation must be multiplication or x and the result is 8.*

Example1: Query

**Question**: *What is the result of the following mathematical expression? Think step-by-step and give proper reasoning steps first and then given your final answer. The format should be > Reasoning: xxx || Answer: xxx .The Reasoning part should contain reasons to derive the answer and the Answer part should only contain the answer. Your response should stricly follow this format and not just give the answer of the mathematical operation. It's important that you give reasoning before you answer.*
**LMM prediction**: *There are two numbers, 6 and 2. We need to find the result of an operation between them. The question is asking for the result of the operation, which is 12. So the answer is Reasoning: The two numbers are 6 and 2. The operation between them is multiplication. $6 \times 2 = 12$ || Answer: 12*

Figure 22: (Operator Induction Math Reasoning) An example of a 1-shot mathematical reasoning task with Judge Rating: 3 (shown in Figure 23)

Example 1: Judge Response

---

**Judge Response –** ### *Analysis of the Problem*

#### *In-Context Example:*
*The in-context example provides the expression '1 ? 8' and states that the result is **8**. The solution explains that among the possible operations (subtraction, multiplication, addition), the operation that yields the result **8** is **multiplication** ('1 × 8 = 8'). This establishes the pattern that the hidden operation is **multiplication**.*

#### *Query Image:*
*The query image presents the expression '6 ? 2'. Based on the in-context example, we infer that the hidden operation is **multiplication**.*

#### *Candidate Response:*
*The candidate response states:*
  - *  **Reasoning**: The two numbers are 6 and 2. The operation between them is multiplication. $6 \times 2 = 12$.*
  - *  **Answer**: 12*

#### *Reference Answer:*
*The reference answer is also **12**.*

### *Evaluation of the Candidate Response*

  1. ***Correct Identification of the Operation**: - The candidate correctly identifies the operation as **multiplication** based on the in-context example. This is consistent with the pattern established in the in-context example.*

  2. ***Correct Application of the Operation**: - The candidate applies the multiplication operation correctly: $6 \times 2 = 12$.*

  3. ***Final Answer**: - The candidate's final answer is **12**, which matches the reference answer.*

  4. ***Clarity of Reasoning**: - The reasoning provided is clear and logically follows the pattern established in the in-context example. The candidate explicitly states the operation and shows the calculation step.*

  5. ***Format Compliance**: - The candidate response follows the required format: "Reasoning: xxx || Answer: xxx".*

### *Final Verdict*
*The candidate's response is both logically sound and mathematically correct. Both the reasoning and the final answer are accurate, and the response adheres to the required format.*

> ***Rating: 3***

---

Figure 23: (Operator Induction Math Reasoning) The Judge (Qwen2.5-VL-32B) evaluates the response of the LMM in Figure 22 to provide correct rating.

Example 2: Support

**Question**: *What is the result of the following mathematical expression?*
**Answer**: *There are two numbers,* 1 *and* 6*. Performing some mathematical operation gives the answer* 6*. So if we think about subtracting the numbers,* $1 - 6 = -5$*, multiplying the numbers,* $1 \times 6 = 6$*, adding the numbers,* $1 + 6 = 7$*. This implies that the hidden operation must be multiplication or* $\times$ *and the result is* 6

Example 2: Query

**Question**: *What is the result of the following mathematical expression? Think step-by-step and give proper reasoning steps first and then given your final answer. The format should be* > Reasoning: xxx || Answer: xxx *.The Reasoning part should contain reasons to derive the answer and the Answer part should only contain the answer. Your response should stricly follow this format and not just give the answer of the mathematical operation. Its important that you give reasoning before you answer.*
**LMM prediction**: *Reasoning: There are two numbers,* 4 *and* 1*. Performing some mathematical operation gives the answer* 5*. So if we think about adding the numbers,* $4 + 1 = 5$*, subtracting the numbers,* $4 - 1 = 3$*, multiplying the numbers,* $4 \times 1 = 4$*. This implies that the hidden operation must be addition or* + *and the result is* 5*. || Answer:* 5

Figure 24: (Operator Induction Math Reasoning) An example of a 1-shot mathematical reasoning task with Judge Rating: 1 (shown in Figure 25)

**Judge Response –** ### *Analysis of the Problem*
#### *In-Context Example:*
*The in-context example provides the expression '1 ? 6' and the solution states that the result is **6**. The reasoning provided suggests testing different operations:*

- *Subtraction: $1 - 6 = -5$;*
- *Multiplication: $1 \times 6 = 6$;*
- *Addition: $1 + 6 = 7$*

*From these tests, the solution concludes that the hidden operation is **multiplication**, as $1 \times 6 = 6$.*

#### *Query Image:*
*The query image is '4 ? 0'. We need to determine the result of this expression based on the same hidden operation identified in the in-context example.*

#### *Reference Answer:*
*The reference answer provided is **0**.*

#### *Candidate Response:*
*The candidate's reasoning and answer are as follows:*
- *  **Reasoning**: The candidate incorrectly identifies the numbers as **4 and 1** instead of **4 and 0**. They test different operations:*
- *  Addition: $4 + 1 = 5$; Subtraction: $4 - 1 = 3$; Multiplication: $4 \times 1 = 4$*
- *  Conclusion: The hidden operation is **addition**, and the result is **5**.*
- *  **Answer**: The candidate concludes that the result is **5**.*

### *Evaluation of the Candidate Response*

1. *  **Correct Identification of the Hidden Operation**:*
   - *The in-context example clearly indicates that the hidden operation is **multiplication** ($1 \times 6 = 6$).*
   - *However, the candidate incorrectly assumes the hidden operation is **addition** based on their misinterpretation of the numbers.*
2. *  **Correct Interpretation of the Numbers**:*
   - *The query image shows the numbers **4 and 0**, but the candidate mistakenly uses **4 and 1** in their reasoning.*
   - *This error leads to incorrect calculations and conclusions.*
3. *  **Correct Application of the Operation**:*
   - *Even if we overlook the misidentification of the numbers, the candidate's reasoning is flawed because they do not apply the correct hidden operation (multiplication) as indicated by the in-context example.*
4. *  4. **Final Answer**:*
   - *The candidate's final answer is **5**, which is incorrect. The correct answer, based on the reference and the hidden operation of multiplication, should be **0** ($4 \times 0 = 0$).*

### *Rating Justification*
*- The candidate's reasoning is **incorrect** because:*

1. *They misidentify the numbers in the query image.*

2. *They incorrectly deduce the hidden operation as addition instead of multiplication.*

*- The candidate's final answer (**5**) is also **wrong**.*

*Given these errors, the candidate's response does not align with the in-context example or the reference answer. Both the reasoning and the answer are incorrect.*

### *Final Rating:*
*> Rating: 1*

Figure 25: (Operator Induction Math Reasoning) The Judge (Qwen2.5-VL-32B) evaluates the response of the LMM in Figure 24 to provide correct rating.

A.2.10 ATTENTION ENTROPY ANALYSIS

We analyze attention patterns by extracting post-softmax attention scores of the underlying LLM for the LLaVA-ATT-Qwen2.5 7B model on the VL-ICL bench for different shot scenarios. For each generation step, we first re-normalize the extracted attention scores over soft prompt tokens (containing image information) $q_i = \dfrac{a_i}{\sum_{j=1}^{|P|} a_j}$, where $P$ refers to soft prompts. We then calculate the normalized Shannon entropy as,

$$H(Q) = \frac{-\sum_{i=1}^{n} q_i \log q_i}{log(n)}$$

to account for the difference in the number of soft prompts. We average this across all generation steps, LLM attention heads, layers, and examples across different VL-ICL datasets for MAPD with FT $\leq 30$ and our best In-Context$^{PD}$ baseline that uses ICL and report results in Table 15. We see a more uniform distribution of attention scores for MAPD over the fixed set of soft prompts, i.e. 256 and hence the entropy remains consistent even on increasing the number of shots. On the other hand, for In-Context$^{PD}$, as we increase the number of shots, the number of soft prompts increase. For example, 8-shot + 1 query, ICL requires $256 \times (8 + 1) = 2304$ soft prompts. Given this relatively larger context, we see many soft prompt tokens containing very low attention scores leading to a lower value of normalized entropy. This highlights an inherent limitation of the LMM that it is unable to attend to all the soft prompts over longer context lengths.

Table 15: Normalized Attention Entropy on VL-ICL Bench. TTA = Test-Time Adaptation. FT $\leq 30$ = Finetuning with $\leq 30$ gradient steps. ICL = In-Context Learning.

| Training Methods | TTA | 0-S | 1-S | 2-S | 4-S | 8-S |
|---|---|---|---|---|---|---|
| In-Context$^{PD}$ | ICL | **0.84** | 0.75 | 0.63 | 0.51 | 0.45 |
| MAPD | FT$\leq 30$ | **0.84** | **0.81** | **0.84** | **0.80** | **0.81** |

A.2.11 SCALING TO MORE SHOTS

Here, we look into the performance of different prompt distillation methods with finetuning-based test time adaptation for larger number of shots and for 3 tasks from the VL-ICL datasets for LLaVA-ATT-Qwen2.5 7B LMM. We see similar performance gains with the introduction of more shots as shown in Table 11. Both the meta-task learning methods, Multi-Task$^{PD}$ and MAPD perform quite well with MAPD showing outstanding performance for Operator Induction.

Table 16: Operator Induction Results.

| Training Methods | Meta-task | 16-S | 32-S | 64-S |
|---|---|---|---|---|
| NoMeta-task$^{PD}$ | ✗ | 73.3 | 73.3 | 80.0 |
| Model-Avg$^{PD}$ | ✗ | 71.7 | 78.3 | 80.5 |
| In-Context$^{PD}$ | ✓ | 58.3 | 53.3 | 76.7 |
| Multi-Task$^{PD}$ | ✓ | 73.3 | 67.7 | 80.0 |
| MAPD | ✓ | **80.0** | **81.0** | **83.3** |

Table 17: CLEVR Count Induction Results.

| Training Methods | Meta-task | 16-S | 32-S | 64-S |
|---|---|---|---|---|
| NoMeta-task$^{PD}$ | ✗ | 35.5 | 30.0 | 36.5 |
| Model-Avg$^{PD}$ | ✗ | 30.0 | 34.5 | 37.0 |
| In-Context$^{PD}$ | ✓ | 25.5 | 34.5 | 32.5 |
| Multi-Task$^{PD}$ | ✓ | 38.0 | **41.5** | 38.5 |
| MAPD | ✓ | **40.0** | 40.5 | **41.0** |

Table 18: TextOCR Results.

| Training Methods | Meta-task | 16-S | 32-S | 64-S |
|---|---|---|---|---|
| NoMeta-task[PD] | ✗ | 29.0 | 26.5 | 30.5 |
| Model-Avg[PD] | ✗ | 29.0 | 29.5 | 31.5 |
| In-Context[PD] | ✓ | 26.5 | 26.0 | 28.5 |
| Multi-Task[PD] | ✓ | 27.0 | **32.5** | **33.5** |
| MAPD | ✓ | **30.5** | 31.5 | 31.5 |

## A.3 TEST-TIME COMPUTE ANALYSIS FOR ICL VS FT

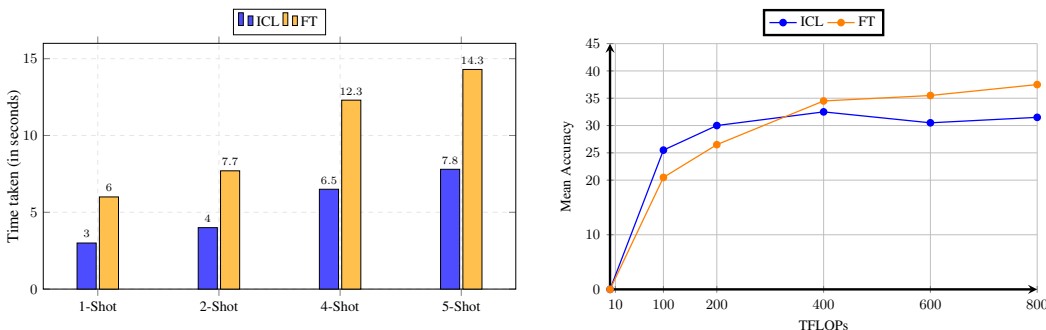

Figure 26: (a) **Left**: Computational time taken per test example (query) by ICL (blue) and FT (orange) for different number of shots. (b) **Right**: FLOPs matched evaluation across all the VL-ICL test sets with mean accuracy for ICL (blue) and FT (orange).

Test-time finetuning (FT) for 30 gradient steps takes about twice as much inference time per test example (query) compared to in-context learning (ICL) under different few-shot scenarios as shown in Figure 26(a). This is not surprising as fine-tuning involves gradient computation, which is more expensive to run than a single forward pass in ICL.

For a more fair comparison, we examine the amount of computation required between these different test-time adaptation methods. Figure 26(b), shows FLOPs-matched evaluation curves for ICL and FT, using In-Context[PD] and MAPD as representative training methods, respectively. We report mean accuracy across all (single-image) VL-ICL datasets. Test-time computation (TFLOPs) scales with the number of shots for ICL, while for FT, it can be scaled by increasing either number of shots or gradient steps. We note that given a low test-time computational budget, ICL performs better than FT, but as the amount of computation is increased FT outperforms ICL. This indicates that FT adaptation is resource-intensive but scales better than ICL as the amount of computation is increased at test time.

After 400 TFLOPs, In-Context[PD] performance begins to decline because the large number of shots used ($\geq$32) exceeds its trained context length of 8,192 tokens. Training In-Context[PD] with longer context would require >4 H200 GPUs, which exceeds our available compute resources. On the other hand, MAPD by design does not require training on long context lengths due to the use of a fixed set of distilled soft prompts for all shots. Additionally, we find that MAPD is much more data-efficient: at 400 TFLOPs, it achieves comparable performance with only 8 shots and 20 gradient steps, indicating better few-shot test-time adaptation.

