# OpenReview forum: "Meta-Adaptive Prompt Distillation for Few-Shot Visual Question Answering"
_ICLR.cc/2026/Conference — ICLR 2026 Poster_

### Official Review · Reviewer_XpJA · 2025-10-28

**Soundness:** 3
**Presentation:** 3
**Contribution:** 2
**Rating:** 6
**Confidence:** 4

**Summary:**

The authors propose a meta-learning–based approach for few-shot adaptation in Large Multimodal Models (LMMs). They observe that training-free in-context learning often yields inconsistent performance in smaller LMMs and fails to scale reliably with more shots. To address this, they introduce learnable soft prompts distilled from image features using a first-order meta-learning framework. Additionally, they replace the standard shallow visual–language projector with a multi-layer self-attention “attention-mapper” module, enabling richer interactions between visual embeddings and prompt tokens before projection into the language space.

**Strengths:**

The paper is well written and clearly explained. It proposes a parameter-efficient approach to fine-tuning Large Multimodal Models (LMMs) using simple yet effective ideas such as learnable soft prompts and meta-learning. The work is timely and highlights an important limitation of standard in-context learning, namely its inconsistent scaling with increasing shots. The proposed method shows strong empirical performance, achieving substantial improvements over standard in-context learning and other baselines on few-shot VQA tasks with LMMs of up to 7B parameters.

**Weaknesses:**

1. Limited technical novelty: The paper borrows heavily from existing approaches. The use of soft or learnable prompts is already well established in few-shot learning and LLM adaptation [1, 2, 3]. Moreover, the meta-learning framework and the attention-mapper module are directly borrowed from Antoniou et al. (2019) and Najdenkoska et al. (2023), respectively.

- [1] Hou et al., MetaPrompting: Learning to Learn Better Prompts, COLING 2022.

- [2] Wang et al., Towards Unified Prompt Tuning for Few-shot Text Classification, EMNLP Findings 2022.

- [3] Khattak et al., Self-Regulating Prompts: Foundational Model Adaptation Without Forgetting, ICCV 2023.

2. Latency and efficiency: Although the method demonstrates strong few-shot performance, one of the main advantages of training-free in-context learning is its low latency. The paper does not provide any analysis or discussion of the computational cost or response-time impact of meta-learning-based adaptation.

3. Insufficient explanation of motivation: The authors claim that additional soft prompts mitigate the issue of overwhelming image embeddings in smaller LMMs, but this is not analyzed beyond aggregate performance results. A visualization of attention patterns could have helped verify whether the prompts actually improve focus. Moreover, it seems counterintuitive that adding more embeddings (via soft learnable prompts) and information would resolve a capacity limitation.

4. Limited scalability evaluation: While the method is tested on multiple LMMs, all models are limited to 7B parameters. It remains unclear how well the approach scales to larger backbones (e.g., 13B or 34B) or whether the benefits persist at that scale.

5. Potential over-claiming in contributions: The contributions section slightly overstates novelty. MAPD largely reuses the first-order meta-learning framework of Antoniou et al. (2019), applied to prompt adapters. Similarly, the authors did not clearly distinguish their flexible adapter design from that of Najdenkoska et al. (2023). Although the authors claim to have proposed the mapper (inspired by Najdenkoska et al.), it appears to be essentially the same module.

**Questions:**

I am looking forward to the authors addressing the weaknesses. In particular, I would appreciate them answering the following questions:

1. How does the proposed attention-mapper differ from the adapter module introduced by Najdenkoska et al. (2023)?
2. How does the meta-learning–based adaptation affect latency or response time of the LMMs compared to standard training-free in-context learning?
3. Does the proposed approach scale to larger LMMs (e.g., 13B or 34B), and do the performance gains persist at that scale?
4. How does adding more soft embeddings (via learnable prompts) help reduce the issue of overwhelming image information in smaller LMMs?

---

> ### Author Response · Authors · 2025-11-21
> **Response to Weakness 1**
>
> We thank the reviewer for highlighting that the paper is well-written and clearly explained. Below, we address remaining concerns with additional experiments and clarifications. **We have also updated the paper with all the new tables, results and explanations and highlighted them in blue.**
>
> > **W1:** Limited technical novelty: The paper borrows heavily from existing....
>
> **R1:** We explain how our work differs below and will incorporate a discussion in the paper:
>
> 1. MetaPrompting by Hou et al. [1] applies MAML-based meta-learning to improve soft prompt initialization for masked language modeling on BERT (0.1B params), evaluating only on few-shot text classification tasks. In contrast, MAPD targets visual question answering in LMMs by meta-learning to distill image embeddings into soft prompts as an alternative to multimodal ICL. We employ first-order MAML with a novel meta-task construction and attention-mapper architecture designed specifically for distilling task-specific visual information in LMMs.
>
> 2. Wang et al. [2] propose unified prompt tuning for adapting BERT to text classification using auxiliary self-supervised prompt selection. Unlike MAPD, they do not employ soft prompts or optimization-based meta-learning for few-shot task adaptation.
>
> 3. Khattak et al. [3] introduce PromptSRC for CLIP-based vision-language encoders, mitigating soft prompt overfitting on image classification tasks by maximizing mutual agreement between vision and language encoders. In contrast, MAPD uses first-order meta-learning for fast test-time adaptation on diverse VQA tasks without overfitting. Additionally, PromptSRC requires both CLIP vision and text encoders, while MAPD uses a CLIP vision encoder paired with an LLM decoder.
>
> Our approach is inspired by the first-order MAML approximation from Antoniou et al. [4] but addresses a fundamentally different problem: few-shot visual question answering. While Antoniou et al. proposed stabilization techniques (multi-step loss, second-to-first order annealing) for training complete models on few-shot image classification, we adopt only their learnable inner-loop learning rates.
>
> We made substantial modifications to optimize only the attention-mapper and soft prompts: integrating the inner-loop SGD optimizer with LLaVA v1.5's outer-loop optimizer and batching meta-tasks sampled across multiple VQA datasets. Our first-order MAML implementation is provided in the Supplementary Material.
>
> > **Q1** How does the proposed attention-mapper differ from the adapter module introduced by Najdenkoska et al. (2023)?
>
> As mentioned in our contributions, we adapt the attention-mapper design from Najdenkoska et al. [5], which implements a permutation-invariant set attention block [6] to learn pairwise similarities across visual features. However, Najdenkoska et al. use only the single [CLS] token from CLIP's vision encoder, limiting the attention-mapper's capacity. Instead, we propose to use the complete set of hidden patch features, enabling the attention-mapper to encode detailed visual information for distillation into soft prompts.
>
> We introduce additional architectural modifications for MAPD: removing dropout layers and scaling to 256 soft prompts, 64× more than Najdenkoska et al. [5], which significantly improves few-shot performance as shown in Table 1 and Figure 5 in the paper.
>
> **We have the updated the paper to appropriately cite the above works in our Related Works Section 2 and have clearly explained the differences.**
>
> [1] Hou et al., MetaPrompting: Learning to Learn Better Prompts, COLING 2022.
>
> [2] Wang et al., Towards Unified Prompt Tuning for Few-shot Text Classification, EMNLP Findings 2022.
>
> [3] Khattak et al., Self-Regulating Prompts: Foundational Model Adaptation Without Forgetting, ICCV 2023.
>
> [4] Antoniou et al. How to train your MAML. ICLR 2019
>
> [5] Najdenkoska et al. Meta Learning to Bridge Vision and Language Models for Multimodal Few-Shot Learning. ICLR 2023
>
> [6] Lee et al. Set Transformer: A Framework for Attention-based Permutation-Invariant Neural Networks. ICML 2019

---

> ### Author Response · Authors · 2025-11-21
> **Response to Weakness 2 and 3**
>
> > **W2:** Latency and efficiency: Although the method demonstrates strong few-shot.....
>
> > **Q2:** How does the meta-learning–based adaptation affect latency....
>
> **R2:** We thank the reviewer for highlighting this and report latency (i.e. computational time taken per test example) for MAPD with FT<=30 gradient steps, comparing with ICL.
>
> **Table R2-1:** *Computational Time for MAPD (in seconds) per test example with the given number of shots. TTA=Test-Time Adaptation, ICL = In-Context Learning, FT<=30 = Test-time Finetuning upto 30 gradient steps*
>
> | Method | 1-shot | 2-shot | 4-shot | 5-shot |
> | --- | --- | --- | --- | --- |
> | TTA = ICL | 3.0 | 4.0 | 6.5 | 7.8 |
> | TTA = FT≤30 | 6.0 | 7.7 | 12.3 | 14.3 |
>
> Test-time fine-tuning for 30 gradient steps approximately doubles inference time per example compared to ICL, as expected due to gradient computation overhead (Updated in Appendix A.3). To enable fair comparison of performance and test-time computation, we conduct FLOPs-matched evaluation in Appendix A.3.
>
> While MAPD with test-time fine-tuning is computationally intensive, it becomes significantly more data-efficient as computation scales. At 400 TFLOPs, MAPD achieves superior performance with only 8 shots and 20 gradient steps compared to ICL using 32 shots (In-Context^PD baseline). ICL is constrained by trained context length and exhibits diminishing returns as shots increase, whereas MAPD with FT<=30 demonstrates monotonically increasing performance. As noted in Section 5, optimizing MAPD's computational efficiency for resource-constrained scenarios remains future work.
>
> > **W3:** Insufficient explanation of motivation: The authors claim that additional soft prompts....
>
> **R3:** We clarify a potential misunderstanding: we do not add soft prompts to in-context examples for the LLM. Instead, soft prompts are prepended to image embeddings for each input image and passed to the attention-mapper, which outputs "distilled" soft prompts. Only these distilled soft prompts are fed to the LLM, it never sees the original image embeddings from in-context examples.
>
> This design addresses the non-monotonic improvement problem when naively prompting LLMs with image embeddings as shots increase. The attention-mapper learns to adapt using few-shot examples at test-time, producing distilled soft prompts that replace both in-context examples and query image embeddings. As illustrated in Figure 3 in the paper, only $H_p$ (distilled soft prompts) serves as LLM input, while $P$ (soft prompts) and $Z_v$ (image embeddings) are inputs to the attention-mapper only.
>
> This approach significantly reduces context length: for example, 8-shot + 1 query, ICL requires $256 × (8+1) = 2304$ image embeddings, while MAPD needs only $256$ embeddings via test-time adaptation. The LLM benefits from this reduced, distilled context, as demonstrated by MAPD's superior performance (Table 1in the paper). Our ablation study (Figure 4a in the paper) shows that scaling soft prompts from $4$ to $256$ improves feature extraction while maintaining substantially lower context length than ICL
>
> Additionally, we analyze attention patterns by extracting post-softmax attention scores on the Operator Induction benchmark for different shot scenarios. For each generation step, we compute the sum of attention scores over soft prompt tokens (containing image information), yielding a ratio between 0 and 1. We average these ratios across all generation steps, LLM attention heads, layers, and examples for MAPD with FT≤30 and our best In-Context^PD baseline that uses ICL. Table R3-1 shows that attention assigned by MAPD is consistent even when increasing the number of shots to a higher value due to it using a fixed set of soft prompts, whereas In-Context^PD suffers from being unable to attend over larger context lengths. We believe one of the major reasons for MAPD’s better few-shot performance is due to its small and fixed context length, even with a large number of shots, which consistently utilizes image information during inference.
>
> **Table R3-1:** *Attention score comparison on soft prompt (image) tokens for Operator Induction benchmark between best test-time finetuning and ICL methods.*
>
> | Model | Method | 1-shot | 2-shot | 4-shot | 8-shot |
> | --- | --- | --- | --- | --- | --- |
> | MAPD | FT<=30 | **0.72** | **0.70** | **0.72** | **0.71** |
> | In-Context^PD | ICL | 0.71 | 0.69 | 0.57 | 0.52 |

---

> ### Author Response · Authors · 2025-11-21
> **Response to Weakness 4, 5 and Question 4**
>
> > **W4:** Limited scalability evaluation: While the method is tested on multiple LMMs....
>
> > **Q3:** Does the proposed approach scale to larger LMMs (e.g., 13B or 34B).....
>
> **R4:** The motivation for designing MAPD, as also mentioned in Section 1 in the paper, is that the ICL performance for LMMs is shown to plateau or even deteriorate as the number of shots is increased. This effect is more often seen in smaller LLMs (<=7B params) as was also noted by Zong et al [1]. Following this, we show that MAPD provides a framework to automatically reduce the context used in ICL by using distilled prompts, and this leads to monotonically increasing few-shot performance. We have shown results that MAPD works best for 3B and 7B models in the paper. Training the attention-mapper with 14B and 34B LLMs exceeds our available computational resources (4 H200 GPUs), so we leave this experiment as future work. Instead, we still extend our results for using a slightly bigger and better Qwen3-8B LLM in Table R4-1 to further verify if this technique scales well. We still see that test-time finetuning outperforms ICL even at a higher scale with an average increase of $18.4$ in accuracy, with MAPD still being the state-of-the-art prompt distillation method. Given the current performance scaling, we believe it should persist at higher scales of 14B and 34B.
>
> **Table R4-1:** *Comparison of prompt distillation methods on Open_MI benchmarks when using Qwen3-8B LLM decoder. We report mean accuracy across 1 to 5 shots with 95% binomial confidence intervals.*
>
> | Methods | ICL | FT≤30 |
> | --- | --- | --- |
> | NoMetaTask^PD | 55.0 ± 0.9 | 72.3 ± 0.9 |
> | ModelAvg^PD | 48.5 ± 0.9 | 69.1 ± 0.7 |
> | In-Context^PD | **63.5 ± 0.7** | 71.4 ± 0.9 |
> | MultiTask^PD | 57.6 ± 0.5 | 80.4 ± 0.6 |
> | MAPD | 60.3 ± 0.5 | **83.5 ± 0.6** |
>
> > **W5:** Potential over-claiming in contributions: The contributions section slightly.....
>
> **R5:** As mentioned in our previous response to W1 and Q1, we borrow the base architecture of the attention-mapper from Najdeskoska et al [2] but make additional changes like using all the vision encoder patch features, removing the dropout layers, and scaling the number of soft prompts, which greatly benefit our approach. We have updated the contributions to better highlight these changes.
>
> > **Q4:** How does adding more soft embeddings (via learnable prompts) help reduce....
>
> **R6:** As clarified in our response to W3, we do not add soft prompts in addition to image embeddings. Instead, the attention-mapper's output -- distilled soft prompts -- replaces both in-context examples and query image embeddings as input to the LLM. These prompts contain task-specific image information but are substantially more compact than ICL.
>
> Figure 4a in the paper demonstrates that MAPD benefits from scaling soft prompts to better extract task-specific features while maintaining low context length (256), avoiding the overwhelming image information problem with a much bigger context in ICL (256 x no. of shots). Figure 3 in the paper illustrates this clearly: the LLM only receives $H_p$ (distilled soft prompts) in MAPD. We will revise the paper for improved clarity.
>
> [1] Zong et al. VL-ICL Bench: The Devil in the Details of Multimodal In-Context Learning. ICLR 2025
>
> [2] Najdenkoska et al. Meta Learning to Bridge Vision and Language Models for Multimodal Few-Shot Learning. ICLR 2023

---

> > ### Author Response · Authors · 2025-11-27
> > **Follow up on Weakness 1 and Question 1**
> >
> > > **W1:** Limited technical novelty: The paper borrows heavily from existing....
> >
> > > **Q1:** How does the proposed attention-mapper differ from the adapter module introduced by Najdenkoska et al. (2023)?
> >
> > **R7:** We follow up here on our response to W1 and Q1 to support our statement. We present the results of an ablation study in Tables R7-1 and R7-2 below for MAPD and MultiTask^PD, respectively, on the VL-ICL datasets. Specifically, in our original implementation, we use all the hidden patch token features from the same layer of the vision encoder as also used by LLaVA v1.5 [1] to encode detailed visual information. We clearly see that simply using the [CLS] representation, as done by Najdenkoska et al. [2], limits the performance of both prompt distillation approaches, and they perform far worse. This validates our hypothesis that using all the hidden patch token features from the vision encoder helps encode detailed visual information, which is beneficial for different VQA tasks.
> >
> >
> > **Table R7-1:** *Ablation study to compare MAPD with test-time finetuning (FT<=30) performance on different VL-ICL datasets when using only [CLS] visual feature v/s using a complete set of hidden patch token features (with 95% binomial confidence intervals)*
> >
> > | Visual features used              | Open-MI       | Operator Induction | CLEVR        | TEXTOCR      |
> > |----------------------|---------------|---------------------|--------------|--------------|
> > | [CLS] token          | 69.6 ± 0.8    | 42.3 ± 0.6          | 23.4 ± 0.2   | 16.6 ± 0.3   |
> > | All Patch tokens (*Ours*)  | **77.9 ± 0.7**    | **47.7 ± 0.5**          | **31.4 ± 0.2**   | **26.4 ± 0.5**   |
> >
> > **Table R7-2:** *Ablation study to compare MultiTask^PD with test-time finetuning (FT<=30) performance on different VL-ICL datasets when using only [CLS] visual feature v/s using a complete set of hidden patch token features (with 95% binomial confidence intervals)*
> >
> > | Method               | Open-MI       | Operator Induction | CLEVR        | TEXTOCR      |
> > |----------------------|---------------|---------------------|--------------|--------------|
> > | [CLS] token          | 63.6 ± 0.8    | 40.3 ± 0.6          | 21.7 ± 0.2   | 13.4 ± 0.3   |
> > | All Patch tokens (*Ours*) | **74.6 ± 0.7**    | **45.1 ± 0.5**          | **29.9 ± 0.2**   | **22.9 ± 0.4**   |
> >
> > [1] Liu et al. Visual Instruction Tuning. NeurIPS 2023
> >
> > [2] Najdenkoska et al. Meta Learning to Bridge Vision and Language Models for Multimodal Few-Shot Learning. ICLR 2023

---

> > > ### Author Response · Authors · 2025-11-27
> > >
> > > Dear Reviewer XpJA,
> > >
> > > As we near the end of the author-reviewer discussion phase, we would like to sincerely thank you for your time and valuable feedback. If there are any remaining questions or points you would like us to clarify, please feel free to let us know. We’re here to support the discussion as best we can. Thank you again for your constructive feedback, which has helped strengthen our paper.
> > >
> > > Best regards,
> > >
> > > The Authors

---

> > > > ### Comment · Reviewer_XpJA · 2025-11-27
> > > >
> > > > I would like to thank the authors for their effort in addressing my concerns. Several points are well clarified, including the distinction from Najdenkoska et al. (2023) and the latency comparison to in-context learning. However, I still feel the overall methodological novelty is incremental, and due to compute constraints it remains unclear how the approach scales to larger LLMs. The motivation is only partially addressed. While my initial misunderstanding is resolved, the use of the term “distillation” is somewhat confusing given that no teacher-student distillation actually occurs.
> > > >
> > > > That said, the overall merit slightly outweighs the remaining concerns, and I still lean toward acceptance. I would suggest the authors make the following adjustments:
> > > >
> > > > 1. Clearly state the contributions to avoid any impression of over-claiming.
> > > > 2. Explicitly outline the limitations associated with increased latency compared to training-free ICL.
> > > > 3. Add a qualitative evaluation in the camera-ready version (if accepted), for example visualizations of soft prompt attention or an information-theoretic measure such as attention entropy.

---

> ### Author Response · Authors · 2025-11-29
> **Our response for Reviewer XpJA (1/2)**
>
> We are grateful that several concerns raised by the reviewer have been clarified, and additional experiments have greatly strengthened the paper. Below, we provide further clarifications on the remaining concerns.
>
> **Q:** *However, I still feel the overall methodological novelty is incremental, and due to compute.....*
>
> **R:** As mentioned in our contributions, this is the first exploration to look into mitigating the issue of few-shot adaptation in LMMs. As initially noted by Zong et al (2015) [1] in Figure 6 of their paper and further verified by us in the updated Introduction Section in Figure 2 in our paper, ICL in LMMs (especially for <=7B params) for image-to-text (I2T) scenarios performs far worse than its text-to-text (T2T) counterparts. This strongly suggests that naively increasing image embeddings in context impairs the model's inherent ICL ability. So, we developed MAPD, a meta-learned prompt distillation approach that keeps the number of image embeddings fixed through the use of a fixed set of soft prompts and an attention-mapper, even when shots increase. For example, 8-shot + 1 query, ICL requires 256 x (8+1) = 2304 image embeddings compared to MAPD with 256 image embeddings. We show that MAPD is the most effective strategy for LMM few-shot adaptation at test-time and monotonically increases performance as shots are increased.
>
> Najdenkoska et al. (2023) [2] developed a meta-learning-based approach solely to avoid reliance on hand-engineered task instruction and focused on smaller LMMs (<500M params). Unlike them, we develop MAPD, based on identifying an important issue of inconsistent ICL performance in larger LMMs (3B to 8B params) (also noted by Zong et al [1]), when shots are increased. Further, our MAPD implementation involves a novel meta-task creation strategy, an attention-mapper module which incorporates all the hidden patch features from the vision encoder, and a first-order MAML with learnable learning rates that offers better stability for training with large models.
>
> Further, we provide a list of our experiments and analysis as a reference to the paper for better clarity below:
>
> **Tables:**
>
> **Table 1 and 11:** MAPD outperforms all other prompt distillation approaches - Multi-Task^{PD} based on multitask learning, In-Context^{PD} based on In-Context Tuning, ModelAvg^{PD} based on model averaging, and NoMetaTask^{PD} based on LLaVA v1.5 Instruction Tuning. We also show that using fine-tuning up to 30 gradient steps for these prompt distillation approaches is a much better method compared to ICL for test-time.
>
> **Table 2 and 12:** MAPD still outperforms other parameter-efficient finetuning strategies like LoRA for few-shot adaptation.
>
> **Table 3:** MAPD is compatible with any LMM-based architecture and different model sizes.
>
> **Table 14:** MAPD is most robust to perturbations in the support images that might hinder few-shot adaptation.
>
> **Table 15:** MAPD maintains its outstanding performance even with a large number of shots, like 32 or 64.
>
> **Figures:**
>
> **Figure 4a:** MAPD with test-time finetuning (FT<=30) scales better as the number of shots and soft prompts are increased compared to In-Context^ with ICL, whose performance deteriorates
>
> **Figure 4b:** MAPD facilitates better task understanding and effectively utilizes the underlying LLM’s reasoning capabilities.
>
> **Figure 5:** Attention-mapper and soft prompts benefit MAPD more compared to other architectures.
>
> **Figure 11:** MAPD achieves better accuracy faster than any other prompt distillation approach
>
> **Figure 16:** Similarity-based few-shot selection benefits test-time finetuning and ICL.
>
> **Figure 26:** Test-time finetuning increases the latency compared to ICL, but is also much more data-efficient.
>
> Apart from this, in response to the concern regarding analyzing attention patterns, we provide **Table R3-1** (listed above in response to the weakness W3). We note that one of the major reasons why MAPD works is that the attention over soft prompt image tokens is consistent even when shots increase compared to ICL, which deteriorates significantly.
>
> **We genuinely believe that our work is novel in terms of the problem statement, methodology, and most importantly, our extensive set of analysis and experimentation, and is a significant contribution to the community. As mentioned in our response to weakness W4 and question Q3, we are limited by the amount of computational resources (4 H200 GPUs) and cannot perform experiments on models larger than 8B. We leave this for future work, depending on the availability of the required computational resources, but still believe from the analysis so far on 3B, 7B, and 8B models that MAPD’s performance scales and maintains the outstanding performance.**
>
> [1] Zong et al. VL-ICL Bench: The Devil in the Details of Multimodal In-Context Learning. ICLR 2025
>
> [2] Najdenkoska et al. Meta Learning to Bridge Vision and Language Models for Multimodal Few-Shot Learning. ICLR 2023

---

> ### Author Response · Authors · 2025-11-29
> **Our response for Reviewer XpJA (2/2)**
>
> **Q:** *While my initial misunderstanding is resolved, the use of the term “distillation” is somewhat confusing given that no teacher-student distillation actually occurs.*
>
> **R:** We acknowledge that our work does not perform the conventional teacher-student distillation but relates to other works that use the term “distillation” in a similar aspect to ours. Honda et al. [1] propose to “distill” the information from many-shot ICL into a concise text-based summary. We, on the other hand, use a concise set of soft prompts that are distilled from the image embeddings. Kang et al. [2] also proposed to use the term “distillation” in their methodology for few-shot image classification and segmentation. They argue that their method is a form of self-distillation in the sense that the ViT-based model is supervised using its own intermediate features. In their approach, they consider the DiNO ViT backbone as the teacher and the full classification-segmentation model as the student, with the only difference being that the student learns additional parameters. Following this self-distillation paradigm, we believe that in our MAPD framework, we think of the frozen vision encoder and LLM as the teacher and the full LMM with the attention-mapper (additional parameters) as the student being trained for few-shot learning. We will update this information in the paper for better clarity.
>
> [1] Honda et al. Distilling Many-Shot In-Context Learning into a Cheat Sheet. EMNLP Findings 2025
>
> [2] Kang et al. Distilling Self-Supervised Vision Transformers for Weakly-Supervised Few-Shot Classification & Segmentation. CVPR 2023
>
> **Q:** *Clearly state the contributions to avoid any impression of over-claiming.*
>
> **R:** As mentioned in our response to your weakness 5, we have updated the contributions in the Introduction section 1 (highlighted in blue) to avoid any impression of over-claiming regarding the attention-mapper for better clarity.
>
> **Q:** Explicitly outline the limitations associated with increased latency compared to training-free ICL.
>
> **R:**  As mentioned in our response to weakness 2, we have highlighted the latency limitation of MAPD in the conclusions section 5 (highlighted in blue) and also included a bar graph comparing the latency of test-time finetuning (FT<=30) and ICL in Appendix A.3.
>
> **Q:** *Add a qualitative evaluation in the camera-ready version (if accepted), for example visualizations of soft prompt attention or an information-theoretic measure such as attention entropy.*
>
> **R:** In Table R3-1 above (in response to weakness 3), the attention analysis of MAPD with test-time finetuning (FT<=30) shows that attention stays consistent over soft prompt image tokens when increasing the number of shots, compared to the ICL baseline where it deteriorates. We will add more examples showing this attention pattern comparison in the camera-ready version.

---

> > ### Author Response · Authors · 2025-12-02
> > **Follow-up for Reviewer XpJA**
> >
> > Following the previous suggestion, we calculated the attention entropy over the soft prompt tokens in Table R8-1 below. Since MAPD uses a fixed set of soft prompts i.e., 256, we see a more uniform distribution of attention weights, hence the entropy remains consistent even on increasing the number of shots. On the other hand, for In-Context$^{PD}$, as we increase the number of shots, the attention weights for soft prompts occurring later in the context become higher than the earlier ones within the context. As the shots increase further, this gap increases, leading to a downfall in entropy. This highlights an inherent limitation of the LMM that it is unable to attend to all the soft prompts over longer context lengths, whereas MAPD does not suffer from this limitation and uniformly attends to its fixed set of soft prompts, which contain task-relevant information.
> >
> > **Table R8-1:** *Entropy calculated on post-softmax attention scores for soft prompt tokens for the Operator Induction benchmark.*
> > | Method | TTA | 0-shot | 1-shot | 2-shot | 4-shot | 8-shot |
> > | :--- | :--- | :--- | :--- | :--- | :--- | :--- |
> > | MAPD | (FT $\le$ 30) | **7.99** | **7.95** | **7.99** | **7.93** | **7.93** |
> > | In-Context$^{PD}$ | (ICL) | 7.98 | 7.74 | 6.22 | 4.17 | 1.59 |

---

### Official Review · Reviewer_pV3j · 2025-10-30

**Soundness:** 3
**Presentation:** 3
**Contribution:** 2
**Rating:** 4
**Confidence:** 4

**Summary:**

The paper presents MAPD, a meta-learning framework that enables few-shot adaptation in large multimodal models (LMMs) for visual question answering (VQA) by distilling task-relevant visual features into a fixed set of soft prompts via an attention-mapper integrated into the projection layer. Unlike in-context learning (ICL), which can degrade in performance with more examples in smaller LMMs due to irrelevant image token noise, MAPD leverages first-order MAML to train the attention-mapper and prompts on meta-tasks constructed from support/query splits. At test time, it adapts to new VQA tasks with a few gradient steps on the support set. Evaluated on the VL-ICL Bench, MAPD consistently outperforms ICL and parameter-efficient fine-tuning methods (e.g., LoRA, TPT), with performance improving monotonically as the number of shots increases.

**Strengths:**

1. Novel and effective integration of meta-learning with multimodal prompt distillation: MAPD is the first approach to apply MAML-style bi-level optimization to distill visual features into soft prompts for LMMs.  Its lightweight attention-mapper (~24M trainable parameters) enables efficient task adaptation, achieving state-of-the-art few-shot VQA performance across diverse VL-ICL tasks, while scaling reliably with support set size.
2. Rigorous experimental design with strong baselines and reproducibility: The authors provide comprehensive ablations (Multi-TaskPD, In-ContextPD, NoMeta-taskPD, Model-AvgPD) and leverage public code, datasets, and a standardized benchmark.  The observed monotonic improvement with increasing shots, coupled with substantial gains over ICL and PEFT methods, offers compelling evidence of MAPD’s practical superiority in low-data regimes.

**Weaknesses:**

1. The paper attributes the non-monotonic improvement in ICL performance of small-parameter LMMs to "irrelevant information interference in image embedding," but it does not rule out other contributing factors, such as the inherent limitations of small models or deviations in instruction understanding.
2. While much prior work has focused on optimizing the projection layer of LMMs, the advantages of this paper relative to existing methods appear limited. his study focuses on scenarios involving small models, single images and limited samples. Does this narrow scope limit its applicability? Additionally, the paper claims that the approach can be "easily incorporated into the projection layer of any LMM architecture," but it remains unclear whether it is effective for other architectures, such as variants of models like Qwen3-VL.

**Questions:**

Please refer to weakness

---

> ### Author Response · Authors · 2025-11-21
> **Response to Weakness 1**
>
> We thank the reviewer for their constructive feedback. We are grateful that they find our work novel and contains rigorous experimentation. Below, we address their remaining concerns. **We have also updated the paper with all the new tables, results and explanations and highlighted them in blue.**
>
> > **W1:** The paper attributes the non-monotonic improvement in ICL performance of small-parameter LMMs....
>
> **R1:** We provide additional results to support our hypothesis that multimodal ICL in LMMs is compromised by the irrelevant information present in image embeddings, leading to non-monotonic improvement as the number of shots is increased.
>
> 1. We first refer to the results in Figure 6 by Zong et al. [1], which calculate image-to-text (I2T) ICL performance on CLEVR and Operator Induction benchmarks. They replace images with equivalent text descriptions while maintaining task instructions, then prompt the LMM. Their results reveal significant performance gaps: text-to-text (T2T) ICL not only outperforms image-to-text (I2T) ICL but also shows monotonic improvement with additional shots. This strongly suggests that naively increasing image embeddings in context impairs the model's inherent ICL ability. To validate this finding, we conduct the same analysis on LLaVA-OneVision-7B in Table R1-1, which shows the average accuracy over both benchmarks. In agreement with Zong et al., text-to-text (T2T) ICL consistently outperforms image-to-text ICL and demonstrates monotonic improvement across shots.
>
> **Table R1-1:** *ICL results comparison for LLaVA-OneVision-7B for Image-to-Text (I2T) and Text-to-Text (T2T). TTA= Test-Time Adaptation*
>
> | Model | Method | 0-shot | 1-shot | 2-shot | 4-shot | 8-shot |
> | --- | --- | --- | --- | --- | --- | --- |
> | LLaVA-OV-7B (I2T) | ICL | 18.5 | 42.8 | 42.2 | 37.5 | 34.1 |
> | LLaVA-OV-7B (T2T) | ICL | **25.5** | **50.9** | **59.9** | **66.4** | **68.6** |
>
> 2. We also conduct further analysis where we add an extra (detailed) task instruction (Please see Appendix A.2.1 for task instructions used) to the original image-to-text (I2T) ICL for these models. The results in Table R1-2 below show that accuracy increases compared to original values which use no extra task instruction. But this still does not resolve the decreasing performance as the number of shots increases. This confirms that adding better task instructions is helpful but cannot solve the non-monotonic improvement issue in LMMs.
>
> **Table R1-2:** *ICL results comparison for LLaVA-OneVision-7B for Image-to-Text (I2T) and I2T with detailed task instruction. TTA= Test-Time Adaptation*
>
> | Model | Method | 0-shot | 1-shot | 2-shot | 4-shot | 8-shot |
> | --- | --- | --- | --- | --- | --- | --- |
> | LLaVA-OV-7B (I2T) | ICL | 18.5 | 42.8 | 42.2 | 37.5 | 34.1 |
> | LLaVA-OV-7B (I2T) + Detailed | ICL | **19.3** | **49.3** | **47.4** | **41.2** | **38.6** |
>
> **We have also included both the tables as a line plot in the Introduction Section 1 for more clarity.**
>
> 3. Additionally, we analyze attention patterns by extracting post-softmax attention scores on the Operator Induction benchmark for different shot scenarios. For each generation step, we compute the sum of attention scores over soft prompt tokens (containing image information), yielding a ratio between 0 and 1. We average these ratios across all generation steps, LLM attention heads, layers, and examples for MAPD with FT≤30 and our best In-Context^PD baseline that uses ICL. Table R1-3 shows that attention assigned by MAPD is consistent even when increasing the number of shots to a higher value due to it using a fixed set of soft prompts, whereas In-Context^PD suffers from being unable to attend over larger context lengths. We believe one of the major reasons for MAPD’s better few-shot performance is due to its small and fixed context length, even with a large number of shots, which consistently utilizes image information during inference.
>
> **Table R1-3:** *Attention score comparison on soft prompt (image) tokens for Operator Induction benchmark between best test-time finetuning and ICL methods.*
>
> | Model | Method | 1-shot | 2-shot | 4-shot | 8-shot |
> | --- | --- | --- | --- | --- | --- |
> | MAPD | FT<=30  | **0.72** | **0.70** | **0.72** | **0.71** |
> | In-Context^PD | ICL | 0.71 | 0.69 | 0.57 | 0.52 |
>
> [1] Zong et al. VL-ICL Bench: The Devil in the Details of Multimodal In-Context Learning. ICLR 2025

---

> ### Author Response · Authors · 2025-11-21
> **Response to Weakness 2**
>
> > **W2:** While much prior work has focused on optimizing the projection layer of LMMs...
>
> **R2** To our knowledge, we are the first to explore meta-learning techniques for improving few-shot abilities in LMMs, given that ICL in these models shows non-monotonic improvement as shots increase. We focus on relatively smaller LMMs (<=7B parameters) because this ICL limitation is most pronounced at this scale, even with single images and few in-context samples, as noted previously by Zong et al. [1]. Our computational resources (4 H200 GPUs) preclude evaluating MAPD on larger-scale LMMs. However, this remains a critical problem: many state-of-the-art LMMs, including LLaVA-OneVision-7B, Qwen-VL, and IDEFICS, struggle with single-image VL-ICL tasks under few-shot conditions (see Zong et al. [1] for extensive evidence). We believe that with sufficient computational resources, MAPD can be trained and applied to larger LMMs and multi-image scenarios to enhance their few-shot abilities. We leave this as future work.
>
> [1] Zong et al. VL-ICL Bench: The Devil in the Details of Multimodal In-Context Learning. ICLR 2025
>
> > Additionally, the paper claims that the approach can be "easily.....
>
> Training MAPD involves training an attention-mapper to learn a good parameter initialization for few-shot finetuning at test-time. Since fine-tuning data for Qwen3-VL is not publicly available, we cannot demonstrate MAPD on this model. Instead, we adopt LLaVA v1.5 as our base architecture, a fully open model with accessible training code and datasets, and demonstrate that MAPD achieves state-of-the-art performance compared to other prompt distillation, parameter-efficient fine-tuning, and ICL approaches across VL-ICL benchmarks. To demonstrate MAPD's architectural flexibility, Table 3 shows results across different underlying LLMs and vision encoders, with MAPD consistently outperforming alternative approaches across all configurations. We further include results in Table R2-1 below, when using a better LLM decoder, Qwen3-8B, with our finetuning data mixture (Appendix A.1.1). We still see that test-time finetuning outperforms ICL even at a higher scale, with an average increase of 18.4 in accuracy with MAPD still being the state-of-the-art prompt distillation method
>
> **Table R2-1:** *Comparison of prompt distillation methods on Open_MI benchmarks when using Qwen3-8B LLM decoder. We report mean accuracy across 1 to 5 shots with 95% binomial confidence intervals. ICL=In-Context Learning. FT<=30 = Finetuning up to 30 gradient steps*
>
> | Methods | ICL | FT≤30 |
> | --- | --- | --- |
> | NoMetaTask^PD | 55.0 ± 0.9 | 72.3 ± 0.9 |
> | ModelAvg^PD | 48.5 ± 0.9 | 69.1 ± 0.7 |
> | In-Context^PD | **63.5 ± 0.7** | 71.4 ± 0.9 |
> | MultiTask^PD | 57.6 ± 0.5 | 80.4 ± 0.6 |
> | MAPD | 60.3 ± 0.5 | **83.5 ± 0.6** |

---

> > ### Author Response · Authors · 2025-11-27
> >
> > Dear Reviewer pV3j,
> >
> > As we near the end of the author-reviewer discussion phase, we would like to sincerely thank you for your time and valuable feedback. If there are any remaining questions or points you would like us to clarify, please feel free to let us know. We’re here to support the discussion as best we can. Thank you again for your constructive feedback, which has helped strengthen our paper.
> >
> > Best regards,
> >
> > The Authors

---

### Official Review · Reviewer_6iP1 · 2025-10-31

**Soundness:** 2
**Presentation:** 2
**Contribution:** 2
**Rating:** 2
**Confidence:** 4

**Summary:**

This paper proposes a test-time fine-tuning method to enhance the accuracy of MLLMs in visual question answering (VQA) compared to in-context learning. The proposed method introduces an attention-based mapping network on top of visual tokens that utilizes soft prompt tokens to distill visual information into soft prompts that are then provided to the LLM. The mapping network is pretrained, then fine-tuned on meta-tasks using meta-learning techniques, and finally is fine-tuned for each task during test-time for few-shot VQA. The paper also constructs several alternative methods for fine-tuning the proposed mapping network for comparison with the meta-learning technique. The results show that the proposed method can outperform in-context learning in few-shot VQA.

**Strengths:**

The proposed method is novel, and shows promising improvements on few-shot VQA. The experiments also provide several interesting insights about the role of attention mapper, soft prompts, and various fine-tuning strategies.

**Weaknesses:**

1. The description of how meta tasks are constructed lacks clarity. The paper must provide a clear table describing the number and composition of meta tasks used for fine-tuning and test-time fine-tuning. It is therefore hard to understand how the test accuracies are computed and can be compared.

2. Looking at the Appendix, it seems that different baselines are fine-tuned with different composition of meta-tasks (Appendix A.1.3): e.g., MAPD has 10-10 (support-query), whereas Multi-Task has 5-5, and In-Context has 10-1. No explanation is provided for these choices, and how they might affect the reported results.

3. The choice of bounding the number of test-time fine-tuning iterations (K<=30) seems arbitrary and makes the results of comparisons unreliable. For example, without any other information, it is possible that at K=40, which may take only a few more minutes, another method outperforms MAPD or the ranking of the methods changes completely. Comparing how long it takes each method to reach a certain accuracy level would be a more clear comparison.

4. The paper seems to miss the simple baseline of test-time fine-tuning just the connector (mapper MLP) of other open-source SOTA MLLMs (LLaVA-OV and Qwen-VL). This is critical to show whether the proposed adaptor matters in practice.

5. The results in Table 9 of Appendix show that ICL is still the SOTA on 3 out of the 4 tasks in VL-ICL when used with other MLLMs (LLaVA-OV and Qwen-VL). These results show that the best current method for few-shot VQA is still performing ICL with SOTA MLLMs. This limits the practical relevance of the proposed method.

6. All the reported results lack confidence intervals, so it is unclear how statistically significant the differences are. I suggest reporting binomial confidence intervals for accuracy to clarify this.

7. The paper seem to not report the time and computation over-head of its method.

**Questions:**

1. Can you provide more details about the number of meta-tasks, their composition, and why they differ between different methods?

2. Can you provide a baseline of fine-tuning the mapping network of LLaVA-OV, Qwen-VL?

3. Can you provide confidence intervals for the results?

4. Can you report time overhead of your method compared to ICL?

---

> ### Author Response · Authors · 2025-11-21
> **Response to Weakness 1 and 2**
>
> We thank the reviewer for their constructive feedback, which helped us identify the strengths and weaknesses of our paper, and for acknowledging our responses. Below, we address remaining concerns with additional experiments and clarifications. **We have also updated the paper with all the new tables, results, and explanations, and highlighted them in blue.**
>
> > **W1:** The description of how meta tasks are constructed lacks clarity......
>
> > **Q1:** Can you provide more details about the number of meta-tasks, their ...
>
> **R1:** We formally define meta-tasks in Section 3.2 as small subsets of examples randomly sampled from a single VQA dataset ($D^i$) within the training data mixture ($p(D)$). Each meta-task consists of support and query sets, both containing a fixed number of VQA examples (image, question, answer triplets). The support set provides few-shot demonstrations to the model, either as in-context examples or for gradient-based adaptation, depending on the prompt distillation method. The query set is used to optimize the LMM (specifically, the attention-mapper parameters in our case) during fine-tuning, and to evaluate performance during inference. This meta-task construction protocol remains consistent across both the fine-tuning stage and test-time fine-tuning, following the framework established by Zong et al. [1]. We have updated the paper to include new tables describing the number and composition of meta-tasks, and we provide further explanation below.
>
> **During test-time adaptation**, we use the publicly available VL-ICL benchmark [1] code to construct meta-tasks of fixed sizes. VQA examples are randomly sampled from the predefined training and test splits of each dataset. Table 5 in Appendix A.1.2 specifies the number of meta-tasks per test set, which remains constant throughout our evaluation. All results reported in the paper represent the average accuracy computed over the query examples of these meta-tasks, ensuring fair comparison across all prompt distillation methods and shot configurations.
>
> **During the attention-mapper fine-tuning stage**, we split the unbalanced data mixture (Table 4 in the paper) into training and validation sets as described in Appendix A.1.2 and A.1.3, then construct meta-tasks by randomly sampling VQA examples. We treat the support-query composition as a tunable hyperparameter alongside those listed in Table 7 in Appendix A.1.4 and perform a grid search to identify the configuration that minimizes validation loss for each prompt distillation method. Table 6 in Appendix A.1.2 details the optimal support-query compositions, number of meta-tasks, and total number of training and validation examples used for each method.
>
> Additionally, for In-Context^PD, we follow the in-context tuning algorithm of Chen et al. [2], which uses only 1 query example per meta-task during training and yields optimal performance for this prompt distillation method. Note that the validation is done across a different number of support examples for robustness, and the total number of training and validation examples remains constant across all methods to ensure fair comparison, regardless of meta-task composition.
>
> [1] Zong et al. VL-ICL Bench: The Devil in the Details of Multimodal In-Context Learning. ICLR 2025
>
> [2] Chen et al. Meta-learning via Language Model In-context Tuning. ACL 2022
>
> > **W2:** Looking at the Appendix, it seems that different baselines are fine-tuned with....
>
> **R2:** As mentioned previously, we treat support-query composition as a hyperparameter alongside others and perform a grid search across a fixed set of values and within our GPU resource constraints (4 H200 GPUs). For each prompt distillation method, we select the configuration that achieves the lowest validation loss following standard train-val-test procedures. Table 7 (for meta-task methods) and 8 (for non meta-task methods) in Appendix A.1.4 provide details of all hyperparameters for which we performed grid search.
>
> Additionally, we verify that our support-query selection procedure is optimal by comparing it against training with another choice of equal numbers of support (8) and query (8) examples. Results on the Operator Induction benchmark (Table R2-1 below) show that this configuration performs worse than our selected support-query composition across all meta-task-based prompt distillation methods.
>
> **Table R2-1:** *Mean accuracy across shots on the Operator Induction task with 95% binomial confidence intervals (and comparison from original scores in Table 1 in paper) when using 8 support examples and 8 query examples during fine-tuning. ICL=In-Context Learning. FT<=30 = Finetuning up to 30 gradient steps*
>
> | Method | FT≤30 | ICL |
> | --- | --- | --- |
> | MAPD | **45.9 ± 0.6** (-1.8) | 9.5 ± 0.5 (-0.1) |
> | Multi-task^PD | 43.7 ± 0.6 (-1.4) | 9.2 ± 0.5 (-0.8) |
> | In-Context^PD | 29.1 ± 0.5 (-1.8) | **18.4 ± 0.7** (-2.2) |

---

> ### Author Response · Authors · 2025-11-21
> **Response to Weakness 3 and 4**
>
> > **W3:** The choice of bounding the number of test-time fine-tuning iterations (K<=30)....
>
> **R3:** We have updated  the Figure 11 of Appendix A.2.2, where we plot the average test accuracy curves for all the prompt distillation methods across different shots for up to 40 gradient steps. We find that K<=30 is sufficient for all prompt distillation methods to converge and show no further increase in performance. We attribute this to our adaptation procedure training only 24M parameters over a few examples at test-time. In contrast, the LoRA baseline (Table 2 in the paper) fine-tunes 300M LLM parameters and fails to converge within 30 steps, resulting in worse performance. So, we finally conclude that K<=30 test-time fine-tuning steps is an empirically validated choice, not arbitrary.
>
> > **W4:** The paper seems to miss the simple baseline of test-time fine-tuning just the connector....
>
> > **Q2:** Can you provide a baseline of fine-tuning the mapping network....
>
> **R4:** We clarify that test-time fine-tuning of other LMMs does not serve as a baseline for our MAPD performance with LLaVA-ATT-Qwen2.5-7B (see Table 1 in the paper), as direct comparison is inappropriate due to fundamental differences in model/embedding sizes, architectures, and vision-language training data.
>
> Nonetheless, we have updated Table 13 in the paper and present results when we finetune the MLP-based connector at test-time for K<=30 gradient steps for LLaVA v1.5, LLaVA-Next, LLaVA-OV, and Qwen2.5-VL models. We also provide a direct comparison with their corresponding ICL performance.  Test-time fine-tuning of the MLP connector consistently improves over ICL, supporting our hypothesis that these LMMs are overwhelmed by the image embeddings during ICL. Fine-tuning enables the connector to distill task-specific information into image embeddings before prompting the LLM, thereby enhancing few-shot performance.
>
> > “This is critical to show whether the proposed adaptor matters in practice.”
>
> Our attention-mapper is not an "adaptor" but rather a replacement for the MLP-based projection layer in LMMs for few-shot VQA tasks, as demonstrated in our ablation study (Figure 5 in the paper) and explained in Section 4.3. Like MLP projection layers, the attention-mapper requires large-scale fine-tuning and benefits from meta-learned parameter initialization for few-shot VQA, as evidenced by MAPD's performance on VL-ICL (Tables 1, 11, and 12 in the paper).
>
> For LMMs like LLaVA-OV, fine-tuning the attention-mapper requires more GPUs (>=12 H200 GPUs) due to their large-scale data fine-tuning mixture (10.4M vision-language examples) and AnyRes training procedure [1], exceeding our compute resources. Similarly, Qwen-VL models lack publicly available fine-tuning data. Given these constraints, we cannot conduct attention-mapper fine-tuning experiments on these architectures.
>
> Instead, Table 3 in the paper demonstrates that within our compute budget (4 H200 GPUs) and fine-tuning data mixture (Appendix A.1.1), the attention-mapper integrates seamlessly with different LMM architectures (vision encoders and LLMs), and below in Table R4-1, we show that it even scales performance with new LLM decoders like Qwen3-8B. MAPD consistently outperforms other prompt distillation approaches across these configurations.
>
> [1] Li et al. LLaVA-OneVision: Easy Visual Task Transfer.
>
> **Table R4-1:** *Comparison of prompt distillation methods on Open_MI benchmarks when using Qwen3-8B LLM decoder. We report mean accuracy across 1 to 5 shots with 95% binomial confidence intervals. ICL=In-Context Learning. FT<=30 = Finetuning up to 30 gradient steps*
>
> | Methods | ICL | FT≤30 |
> | --- | --- | --- |
> | NoMetaTask^PD | 55.0 ± 0.9 | 72.3 ± 0.9 |
> | ModelAvg^PD | 48.5 ± 0.9 | 69.1 ± 0.7 |
> | In-Context^PD | **63.5 ± 0.7** | 71.4 ± 0.9 |
> | MultiTask^PD | 57.6 ± 0.5 | 80.4 ± 0.6 |
> | MAPD | 60.3 ± 0.5 | **83.5 ± 0.6** |
>
> **We have updated Table 3 in the paper to include the above result.**

---

> ### Author Response · Authors · 2025-11-21
> **Response to Weakness 5, 6 and 7**
>
> > **W5:** The results in Table 9 of the Appendix show that ICL is still the SOTA on 3 out of the 4 tasks....
>
> **R5:** In Table 13 in Appendix A.2.5, we provide ICL performance of publicly available LMMs as a reference, but also note that it is not possible to directly compare different LMMs due to their fundamental differences in model architectures, sizes, and training datasets. LLaVA-OV, Qwen-2-VL, and Qwen-2.5-VL are finetuned on substantially more vision-language data (10.4M examples) and with significantly more parameters. Comparing MAPD's performance, trained on only 1.3M examples with 24M trainable parameters, against these large-scale LMMs is inherently unfair.  Unfortunately, we are limited by our computational resources (4 H200 GPUs) and cannot provide results for MAPD with these LMMs trained at a large scale.
>
> Surprisingly, even with limited attention-mapper training, MAPD with test-time fine-tuning outperforms ICL with the much larger LLaVA-OneVision-72B on Open_MI, surpasses the ICL on the 7B version (trained on much more data) on Operator Induction, and beats both the stronger models Qwen2-VL-Instruct and Qwen2.5-VL-Instruct on CLEVR. These results demonstrate MAPD's promise as an optimal training strategy for inducing few-shot abilities in LMMs and suggest strong potential for extension to large-scale LMMs.
>
> > **W6:** All the reported results lack confidence intervals, so it is unclear how statistically....
>
> > **Q6:** Can you provide confidence intervals for the results?
>
> **R6:**  We thank the reviewer for highlighting this and have updated the paper to include 95% binomial confidence intervals.
>
> > **W7:** The paper seem to not report the time and computation over-head of its method.
>
> > **Q7:** Can you report time overhead of your method compared to ICL?
>
> In Table R7-1 below, we report the time overhead of MAPD with FT<=30 gradient steps and its ICL version. We have also updated this table as a bar graph in Appendix A.3 in the paper.
>
> **Table R7-1:** *Computational Time for MAPD (in seconds) per test example with the given number of shots. TTA=Test-Time Adaptation. ICL = In-Context Learning. FT<=30 = Test-time Finetuning up to 30 gradient steps.*
>
> | Method | 1-shot | 2-shot | 4-shot | 5-shot |
> | --- | --- | --- | --- | --- |
> | TTA = ICL | 3.0 | 4.0 | 6.5 | 7.8 |
> | TTA = FT≤30 | 6.0 | 7.7 | 12.3 | 14.3 |
>
> Test-time finetuning for 30 gradient steps takes about twice as much inference time per query under different few-shot scenarios. This is not surprising as fine-tuning involves gradient computation, which is more expensive to run than a single forward pass in ICL. To enable fairer comparison considering both performance and test-time computation, we conduct FLOPs-matched evaluation in Appendix A.3. While MAPD with test-time fine-tuning is computationally intensive, it becomes significantly more data-efficient as test-time computation scales. At 400 TFLOPs, MAPD requires only 8 shots and 20 gradient steps to outperform ICL with 32 shots using our best in-context baseline, In-Context^PD.
>
> ICL is constrained by trained context length and exhibits degradation in performance as shots increase, whereas MAPD with FT≤30 demonstrates monotonically increasing performance with additional gradient steps. As noted in Section 5, optimizing MAPD's test-time computational efficiency for resource-constrained scenarios remains future work.

---

> > ### Author Response · Authors · 2025-11-27
> >
> > Dear Reviewer 6iP1,
> >
> > As we near the end of the author-reviewer discussion phase, we would like to sincerely thank you for your time and valuable feedback. If there are any remaining questions or points you would like us to clarify, please feel free to let us know. We’re here to support the discussion as best we can. Thank you again for your constructive feedback, which has helped strengthen our paper.
> >
> > Best regards,
> >
> > The Authors

---

### Official Review · Reviewer_WEYe · 2025-10-31

**Soundness:** 3
**Presentation:** 3
**Contribution:** 3
**Rating:** 6
**Confidence:** 5

**Summary:**

This paper proposes a meta learning based approach for adding few shot capabilities to LLMs. The paper gives a convincing argument why this is needed and also provide a comprehensive set of baseline methods to show how the new method compares to those. Claim is that through meta learning the method is not hampered by the problem of others methods that they cannot leverage the longer context with more images. There is an extensive set of experiments both in the paper and in the appendix. State-of-the-art results are obtained.

**Strengths:**

- The paper is very clear and comprehensive. Experiments are well defined and serve a clear purpose.
- New approach to the problem which gives clear improvement in the results.
- Good ablation study to analyze specific aspects of the methods.

**Weaknesses:**

- The paper indicates that existing methods cannot leverage the benefits of having longer contexts. This is clearly demonstrated through experimentation. But some more theoretical and intuition basis for this could be given.
- The method can be applied to any existing method but only shown for a limited number of models. It is also not made explicit what is required for a model to be able to apply your method. You are using prompts and not every model is using exactly the same type of prompts.

**Questions:**

Please see the two elements indicating for weaknesses and reflect on those.

**Details Of Ethics Concerns:**

N.A.

---

> ### Author Response · Authors · 2025-11-21
> **Response to Weakness 1**
>
> We thank the reviewer for highlighting the strengths of our work. **Below, we address remaining concerns with additional experiments and clarifications. We have also updated the paper with all the new tables, results and explanations and highlighted them in blue.**
>
> > **W1:** The paper indicates that existing methods cannot leverage the benefits of having longer contexts...
>
> **R1:** In order to build more intuition of why existing methods cannot leverage the benefits of having longer contexts, we perform additional analysis.
>
> 1. We compare the Image-to-Text (I2T) and Text-to-Text (T2T) performance of LLaVA-OneVision-7B LMM on Operator Induction and CLEVR Count Induction tasks in the updated Figure 2 in the paper, and also provide results below in Table R1-1. Our results reveal significant performance gaps: ICL in T2T outperforms I2T, showing monotonic improvement with additional shots. We also observe a decline in performance, even when adding detailed task instructions (see Appendix A.1.2 for task instructions), which suggests that the underlying LLMs are much better at understanding text over long contexts but fail to do so when they encounter image embeddings. This strongly suggests that naively increasing image embeddings in context impairs the model's inherent ICL ability over long contexts and validates our hypothesis.
>
> **Table R1-1:** *ICL results comparison for LLaVA-OneVision-7B for Image-to-Text (I2T), I2T Detailed, and Text-to-Text (T2T). ICL= In-Context Learning*
>
> | Model | Method | 0-shot | 1-shot | 2-shot | 4-shot | 8-shot |
> | --- | --- | --- | --- | --- | --- | --- |
> | LLaVA-OV-7B (I2T) | ICL | 18.5 | 42.8 | 42.2 | 37.5 | 34.1 |
> | LLaVA-OV-7B (I2T) + Detailed | ICL | 19.3 | 49.3 | 47.4 | 41.2 | 38.6 |
> | LLaVA-OV-7B (T2T) | ICL | **25.5** | **50.9** | **59.9** | **66.4** | **68.6** |
>
> 2. Additionally, we analyze attention patterns by extracting post-softmax attention scores on the Operator Induction benchmark for different shot scenarios. For each generation step, we compute the sum of attention scores over soft prompt tokens (containing image information), yielding a ratio between 0 and 1. We average these ratios across all generation steps, LLM attention heads, layers, and examples for MAPD with FT≤30 and our best In-Context^PD baseline that uses ICL. Table R1-2 shows that attention assigned by MAPD is consistent even when increasing the number of shots to a higher value due to it using a fixed set of soft prompts, whereas In-Context^PD suffers from being unable to attend over larger context lengths. We believe one of the major reasons for MAPD’s better few-shot performance is due to its small and fixed context length, even with a large number of shots, which consistently utilizes image information during inference.
>
> **Table R1-2:** *Attention score comparison on soft prompt (image) tokens for Operator Induction benchmark between best test-time finetuning and ICL methods. ICL = In-Context Learning, FT<=30 Test-time Finetuning up to 30 gradient steps.*
>
> | Model | Method | 1-shot | 2-shot | 4-shot | 8-shot |
> | --- | --- | --- | --- | --- | --- |
> | MAPD| FT<=30 | **0.72** | **0.70** | **0.72** | **0.71** |
> | In-Context^PD | ICL | 0.71 | 0.69 | 0.57 | 0.52 |

---

> ### Author Response · Authors · 2025-11-21
> **Response to Weakness 2**
>
> > **W2:** The method can be applied to any existing method but only shown for a limited number of models....
>
> **R2:** In the paper, we apply our approach to different encoders and LLMs (see ablation in Table 3 in the paper), demonstrating that MAPD remains a state-of-the-art approach for inducing few-shot capabilities. We further include results when using a better LLM decoder, Qwen3 8B, in Table R2-1 below. We still see that test-time finetuning outperforms ICL, with an average increase of 18.4 in accuracy, with MAPD still being the state-of-the-art prompt distillation method.
>
> **Table R2-1:** *Comparison of prompt distillation methods on Open_MI benchmarks when using Qwen3-8B LLM decoder. We report mean accuracy across 1 to 5 shots with 95% binomial confidence intervals. ICL = In-Context Learning, FT<=30 Test-time Finetuning up to 30 gradient steps*
>
> | Methods | ICL | FT≤30 |
> | --- | --- | --- |
> | NoMetaTask^PD | 55.0 ± 0.9 | 72.3 ± 0.9 |
> | ModelAvg^PD | 48.5 ± 0.9 | 69.1 ± 0.7 |
> | In-Context^PD | **63.5 ± 0.7** | 71.4 ± 0.9 |
> | MultiTask^PD | 57.6 ± 0.5 | 80.4 ± 0.6 |
> | MAPD | 60.3 ± 0.5 | **83.5 ± 0.6** |
>
> > “It is also not made explicit what is required for a model to be able to apply your method. You are using prompts and not every model is using exactly the same type of prompts.”
>
> MAPD is designed to enhance few-shot learning in LMMs.  We build upon the LLaVA-v1.5 [1] architecture, which follows a standard pipeline: vision encoder --> projection layer --> LLM decoder. Our attention-mapper replaces the conventional MLP-based projection layer and transforms image embeddings into distilled soft prompts for the LLM. This projection layer is critical for bridging vision and language modalities across LMM architectures.
>
> Many prominent LMMs, including LLaVA variants, Qwen-VL [2], Instruct-BLIP [3], Molmo [4], and Intern-VL [5], share this fundamental pipeline structure. We believe MAPD's attention-mapper and soft prompt generation are architecture-agnostic, as demonstrated by our ablation study in Table 3 in the paper, and can be easily integrated into these models. However, GPU resource constraints prevent us from validating this extensibility experimentally, which we defer to future work.
>
> [1] Liu et al. Visual Instruction Tuning. NeurIPS 2023
>
> [2] Yang et al. Qwen3 Technical Report.
>
> [3] Dai et al. InstructBLIP: Towards General-purpose Vision-Language Models with Instruction Tuning. NeurIPS 2023
>
> [4] Deitke et al. Molmo and PixMo: Open Weights and Open Data for State-of-the-Art Vision-Language Models
>
> [5] Chen et al. InternVL: Scaling up Vision Foundation Models and Aligning for Generic Visual-Linguistic Tasks. CVPR 2024

---

> ### Comment · Reviewer_WEYe · 2025-11-26
>
> Thanks for the clarifications and the additional experiments. It has confirmed my assessment of the paper.

---

### Author Response · Authors · 2025-11-21
**Summary of Responses to Reviewers**

Thanks to all four reviewers for their thoughtful feedback on our submission! As there are some shared comments/questions amongst reviewers, we will address them here and provide additional details in specific responses to each reviewer. We have also updated the paper while addressing the reviewer's concerns.

**Meta-task Construction and Test-time Fine-tuning** We have provided extensive clarification, explaining that we treat support-query composition as a tunable hyperparameter and perform grid search to identify the optimal configuration for each method based on validation loss. We include additional tables detailing meta-task composition, the hyperparameter search space, and clarified that the total number of training and validation examples remains constant across all methods to ensure fair comparison. In addition, we clarified that K<=30 is an empirically validated choice rather than arbitrary, demonstrating through updated figures that all prompt distillation methods converge by this point with no accuracy improvement beyond K=30.

**Computational Cost and Latency** In our response, we provide concrete response time data showing that FT<=30 takes approximately twice as long as in-context learning (ICL) per example, which is expected due to gradient computation overhead. However, we also demonstrate that MAPD becomes significantly more data-efficient as computation scales. With a FLOPs-matched evaluation, we show that at 400 TFLOPs, MAPD achieves superior performance with only 8 shots compared to ICL requiring 32 shots.

**Architectural Generalizability** We demonstrate that MAPD is compatible with different vision encoders and LLMs (in Table 3 in the paper). The rebuttal further presents new results where we finetune the MLP-based connector at test-time for K<=30 gradient steps for LLaVA v1.5, LLaVA-Next, LLaVA-OneVision, and Qwen2.5-VL models, while providing a direct comparison with their corresponding ICL performance. Results on these new models confirm our hypothesis that distillation improves few-shot performance. We also acknowledge that training the attention-mapper with even larger models (14B, 34B) exceeds our available computational resources of 4 H200 GPUs and defer these experiments to future work.

**Statistical Significance and Confidence Intervals** We have updated the paper to include 95% binomial confidence intervals for all main results, demonstrating that MAPD's improvements are indeed statistically significant across all benchmarks.

**Technical Novelty** MAPD differs fundamentally from cited prior work:

1. Unlike MetaPrompting [1] (BERT text classification), it targets visual question answering (VQA) in LMMs with novel meta-task construction

2. Unlike PromptSRC [2] (CLIP mutual agreement), it uses first-order MAML for fast test-time VQA adaptation

3. Our attention-mapper adapts Najdenkoska et al [3] but uses all patch features (not just [CLS] token), removes dropout, and scales to 256 soft prompts, a 64× increase that substantially improves few-shot performance.

[1] Hou et al., MetaPrompting: Learning to Learn Better Prompts, COLING 2022.

[2] Khattak et al., Self-Regulating Prompts: Foundational Model Adaptation Without Forgetting, ICCV 2023.

[3] Najdenkoska et al. Meta Learning to Bridge Vision and Language Models for Multimodal Few-Shot Learning. ICLR 2023

**Theoretical and Intuitive Understanding** We conducted attention pattern analysis showing that MAPD consistently assigns higher attention to soft prompt tokens when increasing the number of shots compared to ICL, suggesting better image focus. We also provided text-to-text ICL comparisons demonstrating that when images are replaced with text descriptions, performance improves monotonically, confirming that image embeddings specifically cause the non-monotonic improvement problem. Importantly, we clarified a key misunderstanding: soft prompts do not add to image embeddings but rather replace both in-context examples and query image embeddings, resulting in substantially reduced context length (256 distilled embeddings vs. 2304 image embeddings for 8-shot ICL).

**Performance Highlights** MAPD addresses the non-monotonic improvement problem in multimodal in-context learning (ICL) for Large Multimodal Models (LMMs). Unlike traditional ICL, which becomes overwhelmed by increasing image embeddings, MAPD uses an attention-mapper to distill task-specific information into compact soft prompts, enabling test-time fine-tuning that scales monotonically with shots. It achieves state-of-the-art results across VL-ICL benchmarks with test-time fine-tuning (FT<=30) (see Table 1 in the paper). It also demonstrates **18.4%** average accuracy improvement over ICL when used with new LLMs like Qwen3-8B.

---

### Comment · Area_Chair_hkeX · 2025-11-23

Dear Reviewers,

The authors have submitted their rebuttal addressing your reviews. Please take the time to:

1. Read the rebuttal carefully
2. Ask clarifying questions if anything remains unclear
3. Update your scores and reviews based on the authors' responses

Please be mindful of timing: If you have follow-up questions for the authors, **post them early enough to give them adequate time to respond** before the discussion period closes on December 3rd.

Your timely engagement is crucial for a fair and thorough review process.

Thank you for your continued effort on this paper.

Best regards,
Area Chair

---

### Author Response · Authors · 2025-12-02
**Summary for Area Chair**

Dear Area Chair,

We acknowledge the recent information leak incident on OpenReview and sincerely appreciate your time and effort in reviewing our paper further. In our final comment, we would like to provide a summary of the rebuttal phase.

In this paper, we introduce a meta-learning based prompt distillation approach, MAPD, that uses a few gradient steps (up to 30) at test time to improve few-shot performance in LMMs. MAPD is designed to address an important issue with LMMs: ICL performance deteriorates as the number of shots increases, as noted by Zong et al [1]. MAPD is shown to outperform ICL by **21.2%** on the VL-ICL datasets and also offers monotonically increasing performance as the number of shots are increased (**Tables 1, 2, 11, and 13**).

[1] Zong et al. VL-ICL Bench: The Devil in the Details of Multimodal In-Context Learning. ICLR 2025

**Preliminary Analysis** -  We show from our preliminary analysis that LMMs offer much better performance with ICL when equivalent text descriptions are used in the few shots as opposed to using images within context (**Figure 1 in the paper, Table R1-1 (Reviewer WEYe) and Table R1-1 and R1-2 (Reviewer pV3j)**). This suggests that naively increasing image embeddings in context overwhelms the LMM and impairs its inherent ICL ability.

**Methodology** -  The above issue motivates us to develop MAPD. It learns an attention-mapper in the projection layer and is used to distill only task-relevant features from image embeddings into a fixed set of soft prompts for the underlying LLM. Both the attention-mapper and soft prompts can be easily adapted at test-time using a few examples and require only a few gradient steps (**Figure 3**).

**Ablations** -  We perform an extensive set of analysis and show that our approach works across different model sizes and architectures (**Table 3**), is benefitted by the use of attention-mapper and soft prompts (**Figure 5**), facilitates better task understanding (**Figure 4(b)**), scales better as the number of shots and soft prompts are increased (**Figure 4(a) and Table 15, 16 and 17**), is most robust to image perturbations (**Table 14**) and uniformly distributes attention over soft prompt tokens (**Table R8-1; Reviewer XpJA**)

**Limitations** -  We have also highlighted our computational resource limitations (4 H200 GPUs) and provided results for models up to 8B in size (**Table 3**). Further, we show that MAPD uses test-time finetuning for up to 30 gradient steps, which increases response time compared to ICL due to extra gradient computation, but it also becomes much more data-efficient as computation (TFLOPs) scales (**Figure 26; Appendix A.3**).

**Reviewer Concerns** -

**Reviewer WEYe** – We have provided a more intuitive understanding based on the difference in text-to-text (T2T) and image-to-text (I2T) ICL (**Table R1-1**) performance in LMMs and additional attention analysis (**Table R1-2**). We have also clarified the model requirements and shown that our approach is flexible for any LMM architecture (**Table 3**). The reviewer has acknowledged our responses.

**Reviewer 6iP1** -  We have provided details on meta-task construction (**Tables 5 and 6**) and the choice of support-query compositions and other hyperparameters used for grid search (**Tables 7 and 8**). We have also updated **Figure 11** to show that 30 gradient steps during test-time finetuning is sufficient for the best results for all methods. We have provided results for test-time finetuning of the mapping network and ICL for publicly available models in **Table 13**. We have updated the paper to report 95% binomial confidence intervals and have provided the computational overhead for MAPD in **Table R7-1 and Appendix A.3**.

**Reviewer pV3j** – We have provided analysis to show differences in performance between text-to-text (T2T) (**Table R1-1**) and image-to-text (I2T) (**Table R1-2**) ICL (with and without detailed task instructions) in LMMs and analyzed attention scores (**Table R1-3**) to rule out other contributing factors. We have provided results on Qwen3-8B LLM in **Table R2-1 and Table 3 in the paper** showing MAPD's architectural flexibility.

**Reviewer XpJA** – We have provided differences from prior works and show that our work is novel in terms of problem statements, methodology, and an extensive set of analyses. We have updated the paper to include this information, along with limitations in **Section 5**, while ensuring no over-claiming in the contributions. We have provided latency and efficiency analysis in **Appendix A.3 and Table R2-1**. We have also provided additional analysis on attention scores and clarified the use of soft prompts in **Table R3-1 and Table R8-1**. As mentioned by the reviewer, several concerns raised before have been clarified.

We believe that these additions have thoroughly addressed the raised concerns and demonstrate the effectiveness of MAPD for multimodal few-shot learning in LMMs.

---

### Meta-Review · Area_Chair_RDZ8 · 2026-01-05

**Summary:**

Overall, the reviewers agree that the problem of improving the performance of small-to-mid LMMs under multi-shot multimodal ICL is important, and the motivation of compressing visual information into fixed-length soft prompts via an attention-mapper with a few-step test-time adaptation is clear.

**The evaluation is relatively comprehensive, with solid ablations and analyses.** Results show significant gains over ICL and multiple prompt-distillation/ PEFT baselines on VL-ICL Bench, especially as the number of shots increases.

There are three main concerns:

1. The methodological novelty may be incremental (soft prompts + first-order MAML + attention-mapper).

2. The motivation (why multi-shot image ICL fails) needs stronger evidence or visualization

3. Experimental fairness and practicality details need justification

There is also a scalability concern: validation is limited to 7–8B models, the generalization to larger backbones and broader architectures remains unclear.

**Reviewer Concerns:**

Authors provide comments for all concerns raised by reviewers. Most of them are well addressed. Attention analysis is given for stronger motivation clarification.

The common concern of  **limited model coverage** is **partially addressed.** The rebuttal clarifies that MAPD can be applied to LMMs with a projection layer.
They also report new results on Table 3 using a Qwen3-8B LLM decoder. Yet, experiments on models > 8B scale remain unvalidated due to computational constraints, and training on Qwen3-VL is missing due to the lack of public fine-tuning data.

The overall novelty is still viewed as incremental by reviewer XpJA, yet the overall contribution seems enough.

---

**Reviewer WEYe (Score: 6)**

1. intuition/theory on why longer multimodal contexts hurt ICL

**Largely Addressed.** The rebuttal provides I2T vs. T2T comparisons and additional ICL attention statistics anaylsis. While not a formal theoretical explanation, these help strengthen the intuition.

---

**Reviewer 6iP1 (Score: 2)**

1. Meta-task composition unclear; fairness across methods

**Addressed.** The rebuttal defines meta-tasks, support–query composition, and adds tables over meta-task counts and the hyperparameter grid search.

2. K≤30 test-time steps seems unfair

**Addressed.** Authors report updated convergence curves (up to 40 steps) and argue that all prompt-distillation methods saturate by 30 steps.

3. Missing baselines (LLaVA-OV, Qwen-VL)

**Addressed.** The rebuttal adds baseline results for more public LMMs (LLaVA variants + Qwen2.5-VL) vs their ICL performance.

4. No confidence intervals; unclear statistical significance; No time/compute overhead reported

**Addressed.**

---

**Reviewer pV3j (Score: 4)**

1. Hypothesis not fully convincing

**Largely addressed.** The rebuttal adds I2T vs T2T comparisons, supporting that the issue is specifically tied to image embeddings rather than only instruction clarity.


---

**Reviewer XpJA (Score: 6)**

1. Novelty incremental

**Partially addressed.** Authors clarify problem novelty and list their contributions. Still, the reviewer may continue to view overall novelty as incremental.

2. Latency/efficiency missing; training-free ICL advantage is low latency.

**Addressed.** They provide explicit timing and FLOPs evaluation, arguing improved data-efficiency as compute scales.

**Reviewer Scores:**

**Reviewer WEYe （possibly remain 6)**

The reviewer wrote, “Thanks for the clarifications and the additional experiments. It has confirmed my assessment of the paper.” This suggests the rebuttal strengthened confidence but did not shift the overall score.


**Reviewer 6iP1 (possibly higher than 2)**
The rebuttal directly responds to most methodological and reporting concerns.


**Reviewer pV3j (possibly higher than 4)**
The rebuttal solves most concerns. However, the main outstanding limitation is still scale (no >8B LMM training)


**Reviewer XpJA (possibly remain 6)**
The reviewer explicitly states that the rebuttal clarified key points, and they still lean towards acceptance.

---

### Decision · Program_Chairs · 2026-01-26

Accept (Poster)